# Autophagy disruption and mitochondrial stress precede photoreceptor necroptosis in multiple mouse models of inherited retinal disorders

Fay Newton [1], Mihail Halachev [1], Linda Nguyen [1], Lisa McKie[1], Pleasantine Mill [1] & Roly Megaw [1,2]

Inherited retinal diseases (IRDs) are a leading cause of blindness worldwide. One of the greatest barriers to developing treatments for IRDs is the heterogeneity of these disorders, with causative mutations identified in over 280 genes. It is therefore a priority to find therapies applicable to a broad range of genetic causes. To do so requires a greater understanding of the common or overlapping molecular pathways that lead to photoreceptor death in IRDs and the molecular processes through which they converge. Here, we characterise the contribution of different cell death mechanisms to photoreceptor degeneration and loss throughout disease progression in humanised mouse models of IRDs. Using single-cell transcriptomics, we identify common transcriptional signatures in degenerating photoreceptors. Further, we show that in genetically and functionally distinct IRD models, common early defects in autophagy and mitochondrial damage exist, triggering photoreceptor cell death by necroptosis in later disease stages. These results suggest that, regardless of the underlying genetic cause, these pathways likely contribute to cell death in IRDs. These insights provide potential therapeutic targets for novel, gene-agnostic treatments for IRDs applicable to the majority of patients.

Inherited retinal disorders (IRDs) are the most common cause of blindness in children and adults of working age, affecting 1 in 1380 people[1], and characterised by death of our light-sensing photoreceptors. Retinitis pigmentosa (RP) is the most common form, resulting in an initial loss of rod photoreceptors (followed by non-autonomous cone photoreceptor death)[2,3], whilst other IRDs, such as cone dystrophies, cone-rod dystrophies and macular dystrophies, preferentially affect cones first. Although the clinical features of IRDs are well characterised, it is a highly genetically heterogeneous disorder with causative mutations identified in >280 genes[4]. Therefore, despite recent advances in gene replacement and genome editing therapies for IRDs[5,6], developing treatments that target common downstream disease mechanisms, and are thus applicable to a broader range of causative alleles in a gene-agnostic manner, is a strategic priority for eye health for all IRD patients.

IRDs are caused by mutations in genes encoding a diverse range of proteins, with roles in phototransduction, rhodopsin cycling, photoreceptor structure and intracellular trafficking[7–9]. Whilst our knowledge of the function of these genes has improved, there is comparatively little understanding of how disruption of their function impacts on cellular processes that in turn lead to photoreceptor death[10]. Dysregulation of autophagy, the lysosome-dependent pathway through which damaged cell contents are degraded, has been implicated in some IRD models[11–13]. Further, some studies have

[1]MRC Human Genetics Unit, Institute of Genetics and Cancer, University of Edinburgh, Crewe Road, Edinburgh EH4 2XU, UK. [2]Princess Alexandra Eye Pavilion, NHS Lothian, Edinburgh EH3 9HA, UK. ✉e-mail: roly.megaw@ed.ac.uk

identified apoptosis as the major cell death pathway in IRDs[14–16], while others demonstrate that alternate forms of regulated cell death such as necroptosis, ferroptosis and parthanatos play significant roles[17–22]. In order to protect photoreceptors in IRDs using drug-based strategies, it will be vital to define the common mechanisms of photoreceptor death and the upstream pathways from which they are triggered.

Here, using single-cell RNA sequencing (scRNAseq), RP mouse models and multiple imaging modalities, we provide evidence supporting a model whereby photoreceptors are subject to increased mitochondrial stress early in disease. Further, we show that this leads to defects in autophagy. Finally, we demonstrate that photoreceptors undergo programmed cell death via the necroptosis pathway and that similar pathways could contribute to cell death in RP, regardless of the underlying genetic cause. These insights provide potential therapeutic targets to treat RP in patients carrying a wide range of causative alleles.

## Results

### Single-cell transcriptomes reveal an *Rpgr* mutant-specific population of macrophages and distinct sub-populations of rod photoreceptors

The retinitis pigmentosa GTPase regulator (*RPGR*) is an alternatively spliced gene, producing both constitutive (RPGR[1–19]) and retinal specific (RPGR[ORF15]) isoforms that function to regulate outer segment disc turnover, with pathogenic mutations causing a spectrum of retinal disease, from RP to cone-rod dystrophy to cone dystrophy[7]. In keeping, we previously generated two novel *Rpgr* mutant mouse lines that replicate human pathogenic mutations and exhibit similar disease kinetics to human patients[23]. *Rpgr[Ex3d8]*, which harbours an 8 base pair deletion in exon 3 (resulting in a premature termination codon in all transcripts, thus is a null allele with no expression), develops a faster retinal degeneration than *Rpgr[ORF15]*, which has a 5 base pair truncation mutation within the C-terminal repetitive domain of the retina-specific isoform (see Megaw et al., 2024[23] for schematic representation). Both lines exhibit retinal stress, indicated by increased GFAP upregulation in Müller glia (Supplementary Fig. 1), prior to significant photoreceptor degeneration at 18 months[23]. These lines thus offer excellent models with which to identify shared pathway changes across the IRD spectrum.

To profile different retinal cell states that occur as photoreceptors degenerate, we performed scRNAseq on retinas of 18-month-old *Rpgr[ORF15]* and *Rpgr[Ex3d8]* mice and wild-type littermates. Cluster assignment using Seurat identified clusters corresponding to the major retinal cell types (Supplementary Figs. 2a and 3a), classified by expression of known markers of each cell type (Supplementary Table 1, Supplementary Figs. 2c, 3c). Comparison of the expression profiles of mutant and wild-type cells in rod photoreceptor clusters showed that genes required for photoreceptor function were downregulated in mutant cells compared to wild type (Supplementary Figs. 2d, 3d), with retinal degenerative diseases being the most highly enriched GO terms (Supplementary Fig. 4b).

In addition, analysis of the *Rpgr[ORF15]* scRNAseq data identified a small cluster, characterised by expression of macrophage markers (Supplementary Figs. 2a, b), exclusively composed of cells from the mutant sample. Immunofluorescent staining of retina cryosections confirmed cells, positive for the pan-macrophage marker F4/80 and the microglial marker P2Y12, invading outer layers of *Rpgr[ORF15]* and *Rpgr[Ex3d8]* retinas at 18 months, something not observed in wild type at this age (Supplementary Figs. 2e, 3e). Whilst double positive cells are likely non-invasive, tissue-resident macrophages, we cannot rule out the possibility that circulating monocyte-derived macrophages also contribute to this cell cluster. It is not clear why this population was not present in the *Rpgr[Ex3d8]* scRNAseq data. However, since immunofluorescence shows there to be few F4/80[high] P2Y12[low] cells in the *Rpgr[Ex3d8]* outer retina, the profile may have been lost during quality control stages.

Interestingly, both scRNAseq experiments identified several sub-clusters of rod photoreceptors (Fig. 1a, Supplementary Figs. 2a, 3a). Whilst cells in all sub-clusters expressed rod photoreceptor markers (Supplementary Figs. 2c, 3c), each sub-cluster showed differential gene expression, suggesting distinct expression profiles. Similar characteristic genes for each of these sub-clusters were identified in both scRNAseq experiments (Supplementary Table 1). Pseudotime analysis using Slingshot showed that these sub-clusters follow a trajectory whereby expression of genes required for phototransduction decreases between adjacent clusters (Fig. 1a, b). These sub-clusters may therefore represent rod photoreceptors performing sub-optimally, at different stages of degeneration. Indeed, in the *Rpgr[ORF15]* experiment, clusters with reduced phototransduction gene expression contained a higher percentage of photoreceptors from the mutant sample (Supplementary Tables 2 and 3; Supplementary Fig. 2b). The presence of wild-type cells in these clusters could indicate age-related functional decline.

Next, we analysed genes that showed increased photoreceptor expression along this disease trajectory. KEGG pathway analysis revealed upregulation of mRNA processing, TNF-α/NF-κB signalling and lysosome biogenesis pathways (Supplementary Fig. 4a) suggesting increased activation of inflammatory pathways and increased requirement for lysosomes, which may indicate increased or dysregulated autophagy. Interestingly, we also found several genes with increased expression in more degenerated clusters have a role in PI3K/AKT signalling (Supplementary Figs. 4c, d), a pathway associated with cell survival and regulation of autophagy. Expression of these genes increased progressively in wild-type photoreceptors from 'early' to 'late' clusters but this upregulation was more pronounced in mutant photoreceptors (Fig. 1c, d). To further interrogate this pathway, we assessed AKT protein activity in retinal lysates of the faster degenerating, *Rpgr[Ex3d8]* model. We observed an increase in active, phosphorylated AKT (pAKT) relative to total AKT in retinal lysates from *Rpgr[Ex3d8]* mutants compared to wild type at 12 months (Fig. 1e, f). Further, mass spectrometry analysis of total retinal lysates from six-month-old *Rpgr[Ex3d8]* mice[23] showed decreased expression of the PI3K/AKT pathway inhibitor PTEN and the downstream effector mTOR, compared to wild-type littermates (Fig. 1g). We therefore sought to characterise dysregulation in pathways governed by PI3K/AKT.

### *Rpgr* mutant photoreceptors accumulate autophagosomes

The PI3K/AKT pathway regulates autophagy, the lysosome-dependent pathway through which damaged cell contents are degraded, which has been implicated in photoreceptor degeneration in some RP models[11–13]. In addition to the dysregulation of PI3K/AKT and lysosomal biogenesis pathways, our scRNAseq data also revealed changes in expression of autophagy pathway genes in *Rpgr[ORF15]* and *Rpgr[Ex3d8]* mutant photoreceptors compared to wild type (Fig. 2a, b). To further explore autophagy regulation in RP, we focused further functional experiments on the faster degenerating *Rpgr[Ex3d8]* model. We performed transmission electron microscopy (TEM), which revealed a significant increase in large, vesicle-like structures in mutant photoreceptor inner segments compared to wild-type controls at 6 and 12 months of age (Fig. 2c, d). This suggested an accumulation of autophagosomes or autolysosomes, indicating a defect in autophagy, and was supported by elevated expression of autophagy marker p62 (also known as SQSTM1) in photoreceptor inner segments at 12 months in our *Rpgr[Ex3d8]* mutant (Fig. 3a, b). Interestingly, this accumulation was not present at 18 months (Fig. 3b). Elevated p62 expression in mutant photoreceptors was confirmed in *Rpgr[Ex3d8]* mutant retinal lysates compared to wild type at 12 months with immunoblotting (Fig. 3c, d), as was phosphorylation of p62 at residue S403 (p-p62) (Fig. 3c, e). Phosphorylation at this residue is known to be essential for the autophagic function of p62[24]. Similarly, there was significant accumulation of the lysosomal protein LAMP1 in *Rpgr[Ex3d8]*

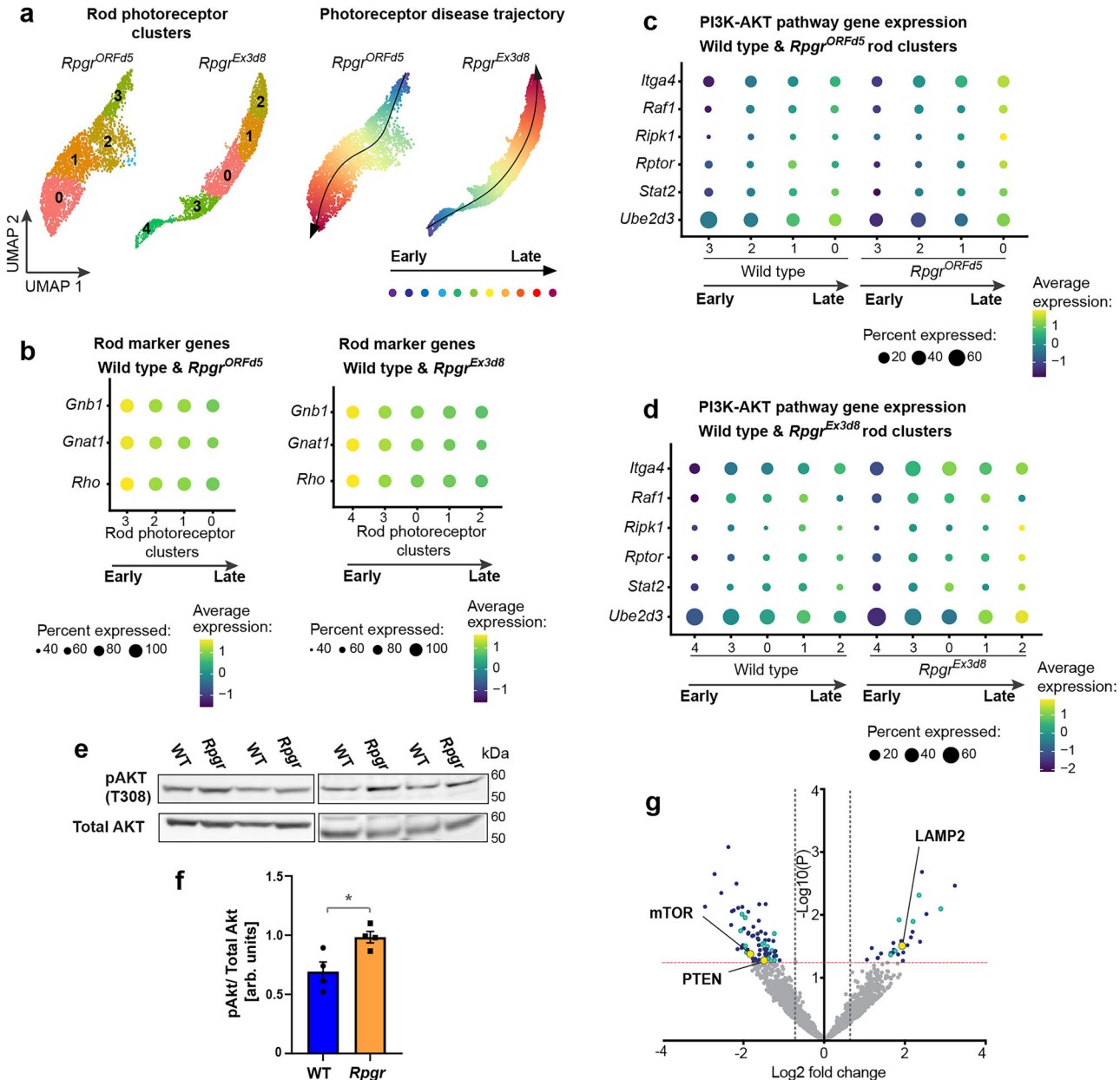

**Fig. 1 | Single-cell transcriptomics identifies photoreceptor populations with reduced phototransduction gene expression and increased PI3K/AKT pathway gene expression. a** Subclusters of rod photoreceptors numbered by Seurat according to cluster size (number of cells in each cluster; left panel). Pseudotime analysis of these clusters shows photoreceptor disease trajectory (right panel). **b** Genes required for phototransduction are downregulated along this trajectory (combined mutant and wild type cells). **c**, **d** Genes associated with PI3K/AKT signalling are upregulated in late (more degenerated) clusters compared to early (less degenerated) clusters. This downregulation is more pronounced in *Rpgr$^{ORFd5}$* (**c**) and *Rpgr$^{Ex3d8}$* (**d**) mutant photoreceptors compared to wild type. **e**, **f** Active pAKT is

increased in *Rpgr$^{Ex3d8}$* mutants relative to total AKT at 12 months (bars show mean; *n* = 4 mice per experimental group; error bars show SEM; *$p$ = 0.037 by unpaired *t*-test). Source data are provided as a Source Data file. **g** Mass spectrometry analysis of *Rpgr$^{Ex3d8}$* retina lysate compared to WT at 6 months (data from Megaw et al., 2024). Red dotted line indicates $p < 0.05$, grey dotted lines indicate Log2 fold change < −0.6 and >0.6. PTEN, mTOR and LAMP2 highlighted by yellow circles, proteins associated with cell stress pathways are highlighted in cyan (listed in Supplementary Table 5). Other proteins with significant fold change (Log2 fold change < −0.6 or >0.6, $p < 0.05$) are highlighted in blue.

photoreceptor inner segments at 12 months, but not at 18 months (Fig. 3f, g), mirroring the increased related protein LAMP2 in our total proteomes of mutant retinas (Fig. 1g). We conclude, therefore, that there is increased autophagy in *Rpgr* mutant photoreceptors in mice up to 12 months of age.

### *Rpgr* mutant photoreceptors exhibit mitochondrial damage
To gain further insight into cellular changes occurring at later stages of degeneration, we carried out TEM on retinas from *Rpgr$^{Ex3d8}$* mice at 12

and 18 months, comparing them to wild-type littermates. At 12 months, when accumulation of autophagosomes was most prevalent in mutant photoreceptors (Fig. 2c), we also observed occasional mitochondrial defects such as swelling or membrane blebbing (Fig. 4a). At 18 months, by contrast, when autophagosome numbers were similar to that of wild type, severe swelling of *Rpgr$^{Ex3d8}$* mitochondria was observed, indicative of mitochondrial stress in mutant photoreceptors (Fig. 4b). Further, mitochondrial genes were upregulated in both *Rpgr$^{ORFd5}$* and *Rpgr$^{Ex3d8}$* mutant photoreceptors in our

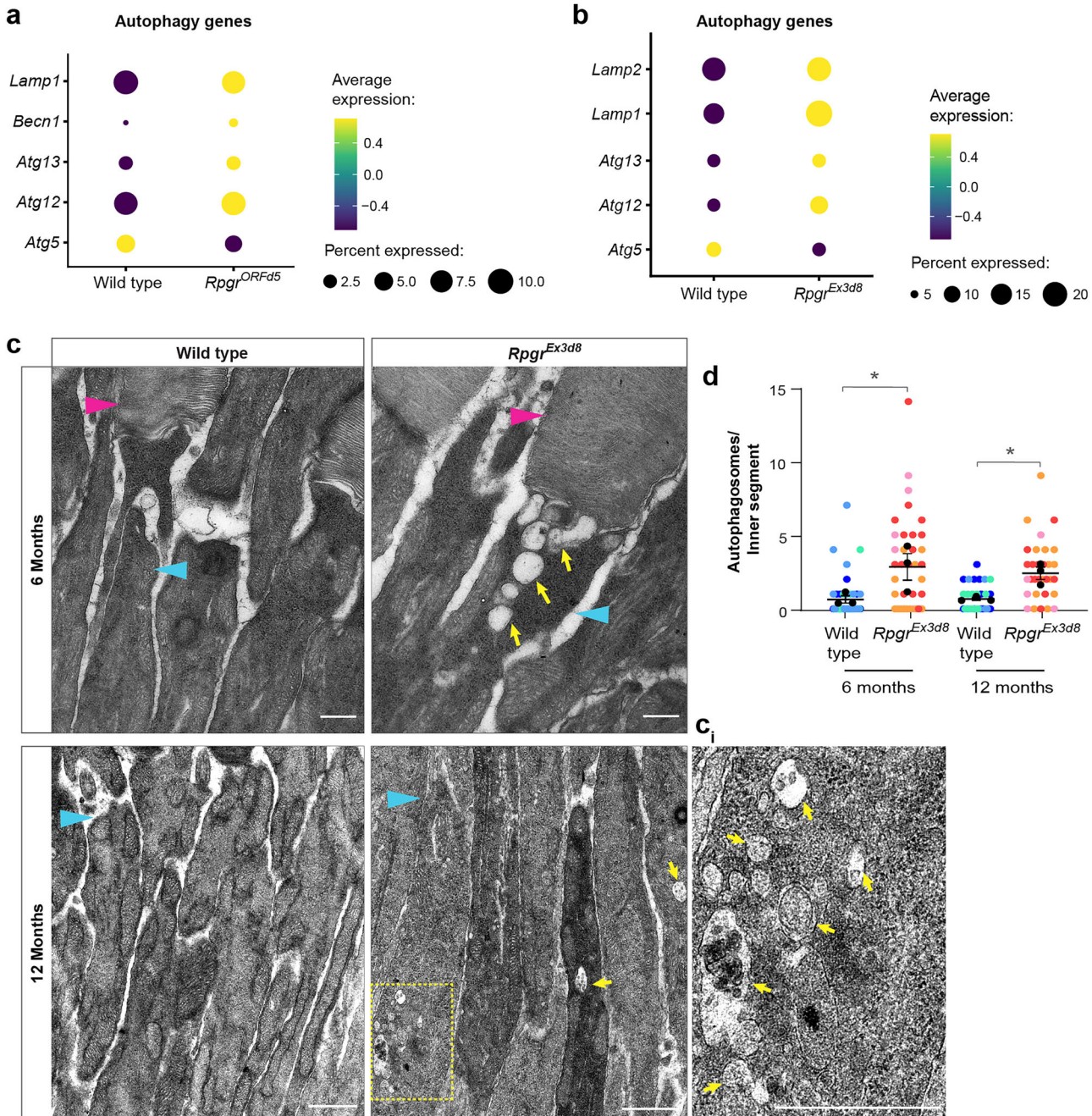

**Fig. 2 | *Rpgr* mutant photoreceptors accumulate autophagosomes in inner segments. a**, **b** Changes in autophagy gene expression in *Rpgr^ORFd5^* (**a**) and *Rpgr^Ex3d8^* mutant (**b**) rod photoreceptors compared to wild type in scRNAseq data. **c** Transmission electron microscopy images of *Rpgr^Ex3d8^* and wild-type littermate control photoreceptor inner segments at 6 and 12 months of age show accumulation of autophagosomes in *Rpgr^Ex3d8^* mutant photoreceptors (magenta arrowheads = outer segments, cyan arrowheads = inner segments, yellow arrows and yellow boxed region indicate autophagosomes, yellow boxed region enlarged in **c_i**, scale bar = 500 nm). **d** Quantification of autophagosome numbers per inner segment. Colours indicate autophagosome counts per photoreceptor from individual mice, with mean values for each mouse superimposed in black (*n* = 3 animals per genotype) error bars show SEM; for each time point, means for individual mutant and wild type animals were compared by unpaired *t*-tests *$p$ = 0.039 (6 months), $p$ = 0.041 (12 months). Source data are provided as a Source Data file.

scRNAseq data (Fig. 4c, d; Supplementary Table 4). Mitochondrial biogenesis genes were also elevated in both *Rpgr* mutant scRNAseq datasets (Supplementary Fig. 5a, b), suggesting a compensatory response to replenish the pool of damaged mitochondria. We further confirmed by immunoblot that some proteins required for oxidative phosphorylation were increased in *Rpgr^Ex3d8^* retinal lysates at 12 months (Supplementary Fig. 5c, d). Interestingly, these were components of the latter part of the electron transport chain (cytochrome C (CYTC) and the complex IV protein cytochrome oxidase 1 (mtCO1)), which

could reflect an attempt to enhance ATP production efficiency or to minimise reactive oxygen species (ROS) production under stress conditions.

Our scRNAseq data also showed upregulation of *Vdac1*, a gene encoding a mitochondrial outer membrane (MOM) calcium channel protein with a role in the mitochondrial stress response[24]. VDAC1 forms part of the MOM complex required for crosstalk with the endoplasmic reticulum (ER) and regulates apoptosis, autophagy and mitophagy[25,26]. Under stress conditions, VDAC1 is upregulated and

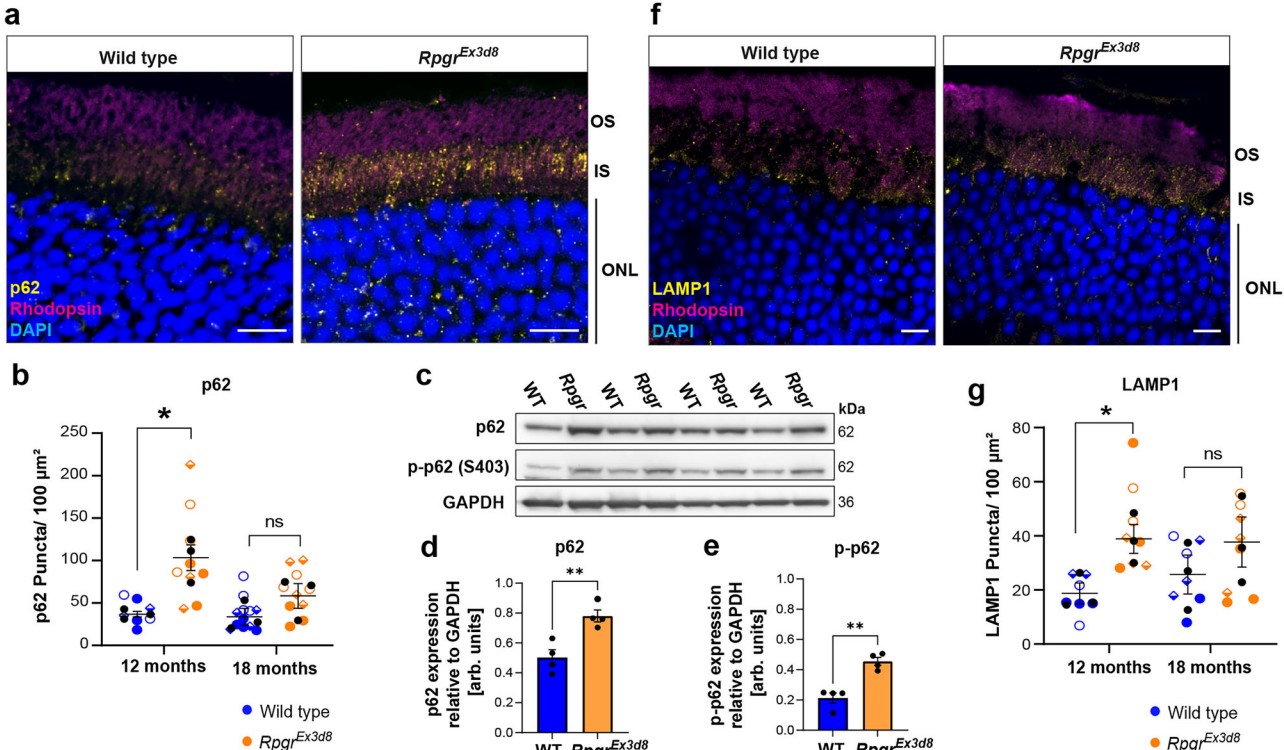

**Fig. 3 | Autophagosome and lysosome markers accumulate in *Rpgr* mutant photoreceptor inner segments. a** Accumulation of p62 in *Rpgr^Ex3d8^* photoreceptor inner segments at 12 months (scale bar = 10 μm). **b** Quantification of p62 puncta in photoreceptor inner segment (*n* = 3 animals per genotype; symbols indicate sections measured from the same animal, means for each animal overlaid in black circles; error bars show SEM; for each time point, means for individual mutant and wild type animals were compared by unpaired *t*-tests *\*p* = 0.043). **c, d** Levels of p62 protein are also increased in *Rpgr^Ex3d8^* whole retina lysates compared to wild type at 12 months (bars show mean; *n* = 4 mice per experimental group; error bars show

SEM; *\*p* = 0.006 by unpaired *t*-test). Levels of autophagy-activating S403 phosphorylation are also increased in mutants (**c, e**) (bars show mean; *N* = 4 mice per experimental group; error bars show SEM; *\*\*p* = 0.002). **f** Accumulation of LAMP1 in *Rpgr^Ex3d8^* photoreceptor inner segments at 12 months (scale bar = 10 μm). **g** Quantification of LAMP1 puncta in inner segment (*n* = 3 animals per genotype, means for each animal overlaid in black circles; error bars show SEM; for each time point, means for individual mutant and wild type animals were compared by unpaired *t*-tests *\*p* = 0.043). Source data are provided as a Source Data file.

forms oligomers at the MOM, increasing membrane permeability and promoting cytochrome C release[27,28]. Increased levels of VDAC1 in *Rpgr^Ex3d8^* mutant photoreceptor inner segments by immunofluorescence at 12 and 18 months further supports an increase in mitochondrial stress (Fig. 5a–c).

To directly assess mitochondrial function in *Rpgr* mutants, we measured oxygen consumption rate (OCR) in ex-vivo retinal punches from 14 month-old *Rpgr^Ex3d8^* mice and wild-type littermates. We observed a significant reduction in basal OCR in *Rpgr^Ex3d8^* retinas, suggesting reduced mitochondrial respiration (Fig. 5d, e). We found no difference between basal and maximal OCR in either mutant or wild-type retinas after adding an uncoupler (FCCP) and therefore could not measure mitochondrial reserve capacity. However, this is consistent with previous studies showing that, due to high oxygen demand, photoreceptor mitochondria function close to maximum capacity[29]. Taken together, these findings support that *Rpgr^Ex3d8^* mutant photoreceptors develop mitochondrial stress from 12 months, worsening as the retina degenerates.

## Necroptosis is the major cell death pathway in *Rpgr* photoreceptors

Mitochondrial stress is associated with increased apoptosis, due to release of cytochrome C from mitochondria[27]. To determine whether apoptosis contributes to the high level of photoreceptor death in *Rpgr* mutants at later stages, 18 month retinal sections were stained with antibodies for cleaved caspase 3 (cl-casp3). We observed few but increased cl-casp3 positive photoreceptors in mutant retinas

compared to wild type (Supplementary Fig. 6a). This relatively small number of positive cells seemed insufficient to explain the significant loss of photoreceptors observed at this age[23] and analysis of our scRNAseq revealed no obvious apoptotic transcriptional signature (Supplementary Figs. 6b, c). This suggested that apoptosis may not be the primary mechanism of photoreceptor death in *Rpgr* mutants.

*Rpgr^Ex3d8^* mutant retinas have reduced cleaved caspase 8 (cl-casp8) relative to the full-length protein (fl-casp8, Supplementary Figs. 6d, e), suggesting less catalytic activity and a subsequent reduction in its extrinsic activation of apoptosis. Further, catalytically active caspase 8 regulates the balance between cell death mechanisms by inhibiting necroptosis[30–32], a form of regulated cell death characterised by high inflammation and cell membrane rupture. In keeping, gene ontology differential expression analysis of our scRNAseq data suggested that both *Rpgr^ORFd5^* and *Rpgr^Ex3d8^* mutant photoreceptors had upregulation of necroptosis-associated genes (Fig. 6a, b, Supplementary Fig. 4c). Necroptosis can also be induced by mitochondrial stress[33]. We therefore investigated whether necroptosis could be involved in photoreceptor death in *Rpgr* mutants.

Activation of necroptosis can happen through a variety of upstream signals, culminating in the phosphorylation and oligomerisation of the pseudokinase Mixed Lineage Kinase Domain-Like protein (MLKL). MLKL oligomers trigger cell membrane destabilisation and consequent pore formation during necroptosis, allowing the release of cellular contents[34]. Key to the execution of necroptosis is the activation of MLKL through RIPK3-mediated phosphorylation of serine 345 (S345) in murine MLKL[34], which can be monitored in situ in genetically

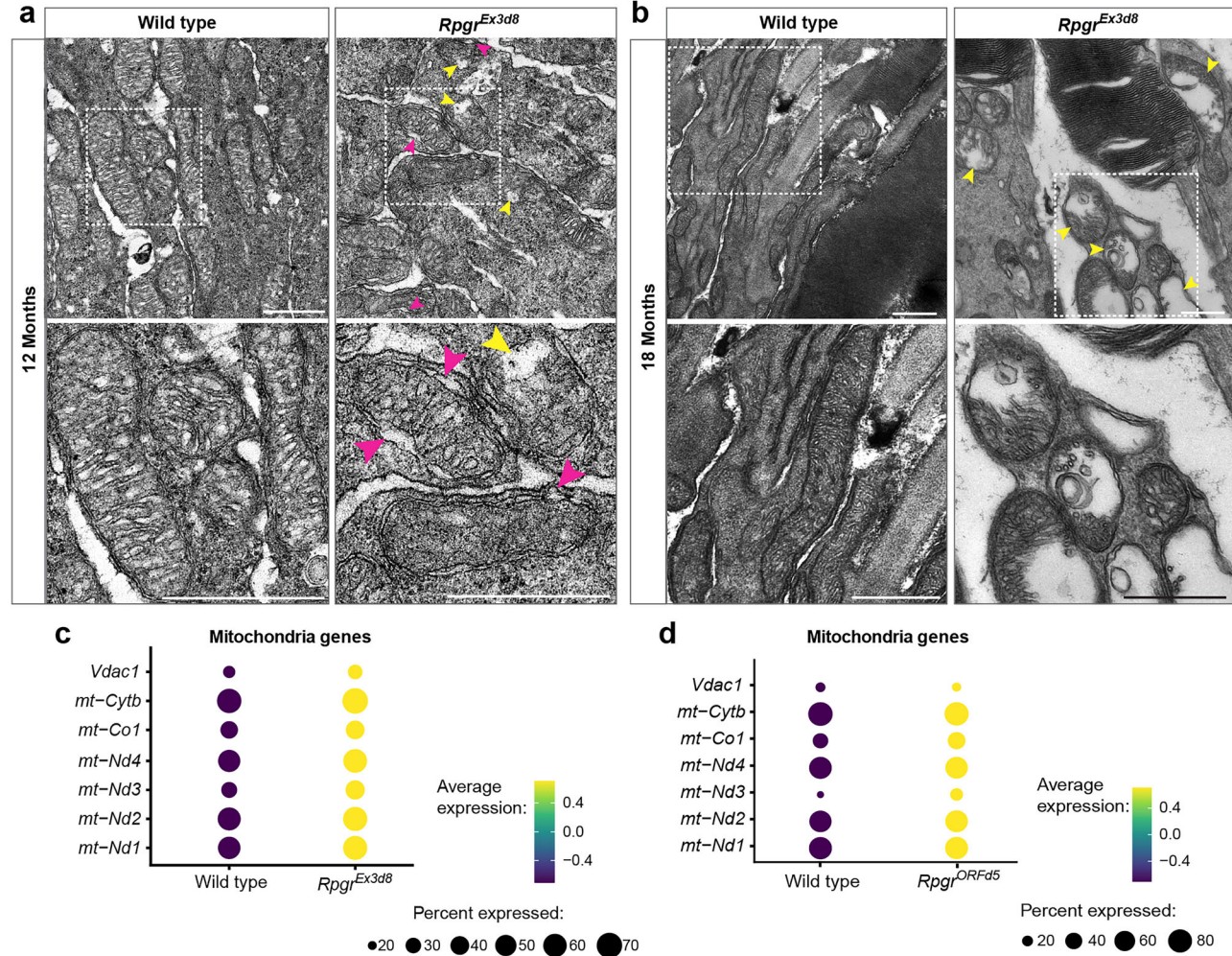

**Fig. 4 | *Rpgr*-mutant photoreceptors have abnormal mitochondrial morphology, indicative of mitochondrial damage. a, b** Transmission electron microscopy images of mutant and wild-type photoreceptor inner segments at 12 months (**a**) and 18 months (**b**) show mitochondrial swelling (yellow arrowheads) and blebbing of mitochondrial membranes (magenta arrowheads) at 12 months in mutant retina ($n = 3$ animals for each genotype at each time point). Boxed regions enlarged in lower panels (scale bar = 500 nm). **c, d** Increased expression of genes required for mitochondrial function in *Rpgr*$^{Ex3d8}$(**c**) and *Rpgr*$^{ORFd5}$ (**d**) mutant photoreceptors in scRNAseq data.

defined models of necroptosis by immunofluorescnce[35,36]. In *Rpgr*$^{Ex3d8}$ and *Rpgr*$^{ORFd5}$ mutant retinas, immunofluorescence revealed increased pMLKL positive photoreceptors, with pMLKL accumulating at the membrane around the cell body (Fig. 6c, d; Supplementary Fig. 7). A much larger number of photoreceptors were affected compared to the number of photoreceptors positive for cl-casp3 at 18 months (Fig. 6d, Supplementary Fig. 6a), suggesting necroptosis is the major cell death pathway in *Rpgr* mutant models of RP. The number of pMLKL positive cells also increased progressively from 6 to 18 months (Fig. 6d, Supplementary Fig. 7b), corresponding to the gradual loss of photoreceptors over this time period. Together, these data suggest necroptosis is the predominant cell death pathway in *Rpgr*-mutant photoreceptors.

**Necroptosis, autophagy disruption and mitochondrial stress also contribute to photoreceptor degeneration in a *Pde6b* mouse model of RP**

To test whether these pathway changes also play important roles in other RP models, we examined the *Pde6b*$^{atrd2}$ mouse[37]. In contrast to RPGR's role in photoreceptor outer segment maintenance, the PDE6 protein complex is involved in the phototransduction cascade and thus represents a suitably distinct model with which to probe for gene-

agnostic, shared pathway disruptions. *Pde6b* mutations result in much faster photoreceptor degeneration, with the *Pde6b*$^{atrd2}$ mouse showing extreme thinning of the ONL by P28[37]. pMLKL expression was significantly higher in *Pde6b*$^{atrd2}$ photoreceptors and increased progressively as they die, from P14 to P28 (Fig. 7a, b). As observed in *Rpgr*$^{Ex3d8}$, levels of active pAKT were increased in *Pde6b*$^{atrd2}$ mutant retinal lysates relative to total AKT at P16 (Fig. 7c, d) suggesting that early upregulation of the PI3K/AKT pathway may be common to both RP models. We also observed increased prevalence of enlarged autophagosomes at P13 (Fig. 7e, f). Furthermore, an accumulation of p62 in *Pde6b*$^{atrd2}$ photoreceptor inner segments was observed at P14 and P16 (Fig. 7g, h) with total increased levels of both p62 and p-p62 (S403) in P16 retinal lysates (Fig. 7i–k).

Progressive defects in mitochondrial morphology from P13-P18 were apparent in *Pde6b*$^{atrd2}$ photoreceptors, similar to those observed in older *Rpgr* mutants (Fig. 8a). As with *Rpgr*$^{Ex3d8}$ mutants, a greater number of VDAC1 puncta were present in *Pde6b*$^{atrd2}$ photoreceptor inner segments compared to wild type at P14 (Fig. 8b, c), suggesting an upregulation of VDAC1 in response to mitochondrial stress. Interestingly, the same oxidative phosphorylation proteins found to be upregulated in *Rpgr*$^{Ex3d8}$ mutant retinas at 12 months (CYTC and mtCO1) were also upregulated in *Pde6b*$^{atrd2}$ mutants at a late disease stage

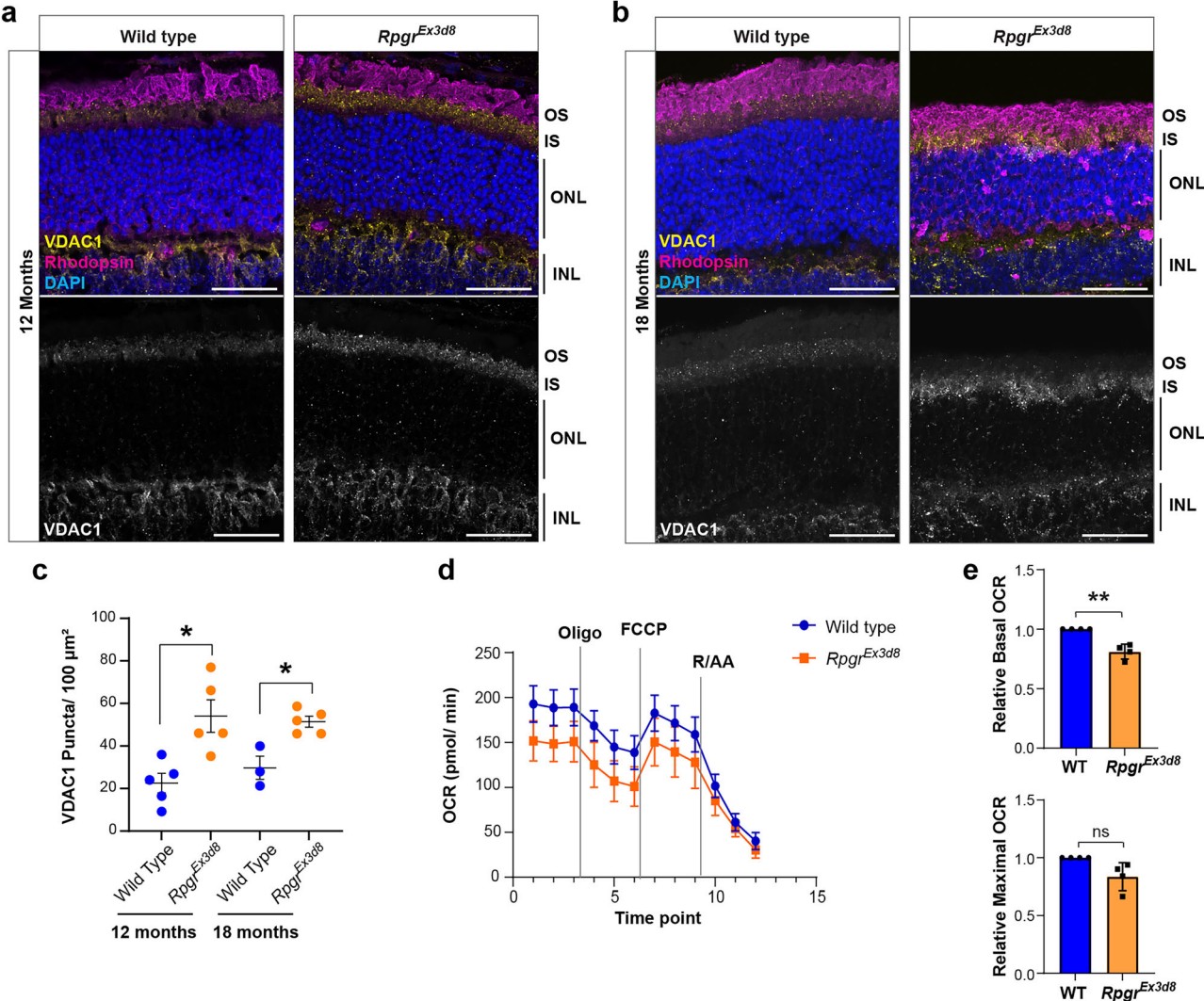

**Fig. 5 | Mitochondrial stress develops in *Rpgr* mutant photoreceptors.**
**a**, **b** VDAC1 expression is increased in mutant photoreceptor inner segments at 12 (**a**) and 18 (**b**) months compared to wild-type littermate controls (scale bar = 25 μm. **c** Quantification of VDAC1 puncta at 12 and 18 months ($N = 5$ animals per genotype, except wild type at 18 months where $n = 3$ animals); error bars show SEM; *$p = 0.01$ (12 months), $p = 0.039$ (18 months) by unpaired *t*-test. **d** Mitochondrial stress assay shows basal oxygen consumption rate (OCR) is reduced in *Rpgr*$^{Ex3d8}$ mutants at

14 months. Mean OCR for 4 biological replicates ($n = 4$ animals per genotype; 10 biopsies per animal) is shown at each time point (error bars show SD). Vertical lines indicate points of addition of relevant modulators (oligo oligomycin, FCCP carbonyl cyanide p-trifluoro-methoxyphenyl hydrazone; R/AA rotenone + antimycin A). **e** Mean OCR of mutant retina (time point 3 = Basal, time point 7 = Max) normalised to the mean OCR of wild-type littermate controls at the same time point (error bars show SD; **$p = 0.009$ by *t*-test). Source data are provided as a Source Data file.

(Supplementary Fig. 5e, f). We conclude, therefore, that dysregulation of autophagy, mitochondrial stress and necroptosis are the key mechanisms contributing to photoreceptor death in multiple RP models, regardless of the underlying cause and gene function (Supplementary Fig. 8).

## Discussion

Understanding the processes underlying photoreceptor degeneration in RP remains an important challenge for the development of effective therapies. Here, we identify mitochondrial stress and dysregulated autophagy as major contributors to photoreceptor deterioration in *Rpgr* and *Pde6b*$^{atrd2}$ mutants, with most cells eventually dying by necroptosis (Supplementary Fig. 8). In contrast to apoptosis, necroptosis is a form of regulated necrosis involving cellular membrane rupture, release of damage associated molecular patterns (DAMPs) and, consequently, a proinflammatory environment resulting in damage to surrounding tissues[38,39]. This is in keeping with the clinical picture of RP, where a chronic, low-grade inflammation often leads to

posterior, sub-capsular cataract, epiretinal membrane formation and cystoid macular oedema.

We found that PI3K/AKT signalling is upregulated in *Rpgr* and *Pde6b*$^{atrd2}$ mutant photoreceptors. PI3K/AKT signalling is generally associated with cell survival; activation of AKT inhibits apoptosis[40–42] and could therefore represent a protective mechanism in response to cell stress. However, sustained activation of AKT triggers necroptosis in neuronal cells[43] and could play a similar role in *Rpgr* mutant photoreceptors. Several other studies, using different models of RP, have identified necroptosis as the primary mechanism of photoreceptor death[17,18,20]. Our finding, therefore, that necroptosis has a significant role in photoreceptor loss in multiple RP mouse models supports that this mechanism may be important in RP, regardless of the genetic cause, and its inhibition could be an attractive target for development of future therapies.

mTOR, reduced in our mutant retinas, acts to inhibit autophagy[42,44]. Depletion of mTOR, therefore, could explain the increased autophagy observed in *Rpgr* mutant photoreceptors at early

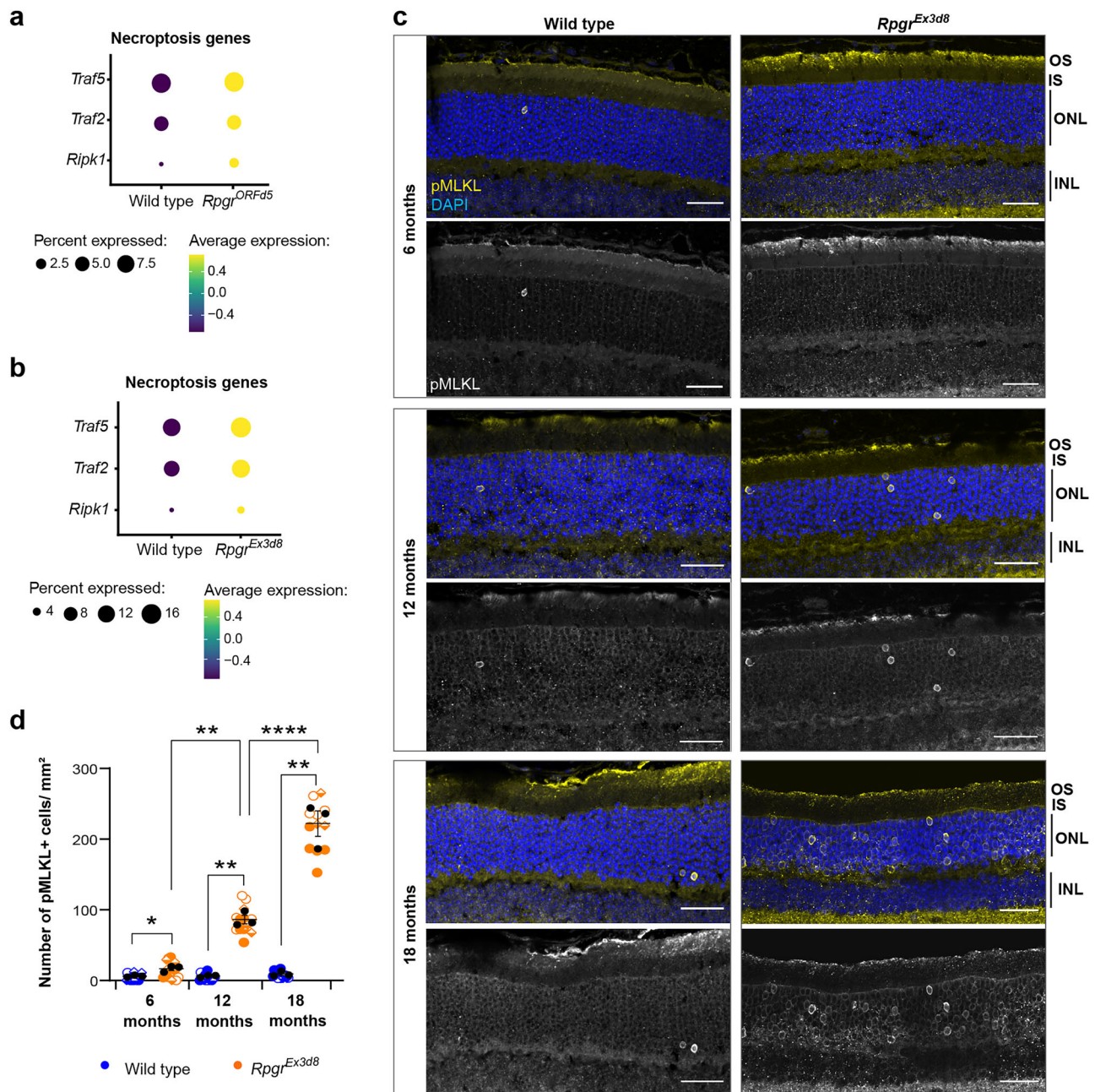

**Fig. 6 | Photoreceptors die by necroptosis in *Rpgr* mutants. a, b** Necroptosis genes have increased expression in *Rpgr^ORFd5* (**a**) and *Rpgr^Ex3d8* (**b**) mutant rod photoreceptors in scRNAseq. **c** Increased pMLKL positive photoreceptors in *Rpgr^Ex3d8* mutants at 6, 12 and 18 months compared to wild-type littermate controls (all scale bars = 25 µm). **d** Quantification of pMLKL positive photoreceptors per mm² in each retina section (symbols indicate images from individual mice, means for each animal overlaid in black circles ($n = 3$ animals per genotype); bars show mean; error bars show SEM; for each time point, means for individual mutant and wild type animals were compared by unpaired $t$-tests with Welch's correction *$p = 0.041$; **$p = 0.005$ (12 months), $p = 0.007$ (18 months). Means for each animal were compared across time points by 2-way ANOVA**$p = 0.003$ (6 versus 12 months); ****$p = 7.63E^{-7}$ (12 vs 18 months)). Source data are provided as a Source Data file.

disease stages. We also observed increased autophagy in *Pde6b^atrd2* photoreceptors at early disease stage, in keeping with other *Pde6b* models[45], suggesting dysregulation of autophagy could play an important role in early photoreceptor degeneration arising from different genetic causes. This increase in autophagy may be a protective response to remove misfolded or mis-localised proteins such as rhodopsin (Supplementary Fig. 1). It could also be an attempt to remove the damaged mitochondria seen in our mouse models, which would be in keeping with the increased oxidative stress seen in our mutant scRNAseq and total proteome datasets. Indeed, metabolic and oxidative stress appear to play significant roles in photoreceptor

degeneration in several other models of RP[46–50]. Further work is needed to resolve the content of the observed autophagosomes. However, photoreceptor autophagy is complex, and inhibition of autophagy appears protective in certain mouse RP models[12].

Alternatively, the autophagosome and lysosome accumulation we observed could indicate progressively defective autophagy, with stalled autophagosomes failing to clear stressed mitochondria and triggering cell death. This could explain why significant accumulation of p62 and LAMP1 was not observed at 18 months, when severely affected photoreceptors have been lost. Photoreceptors have a high basal level of autophagy[51] and may therefore be particularly sensitive to

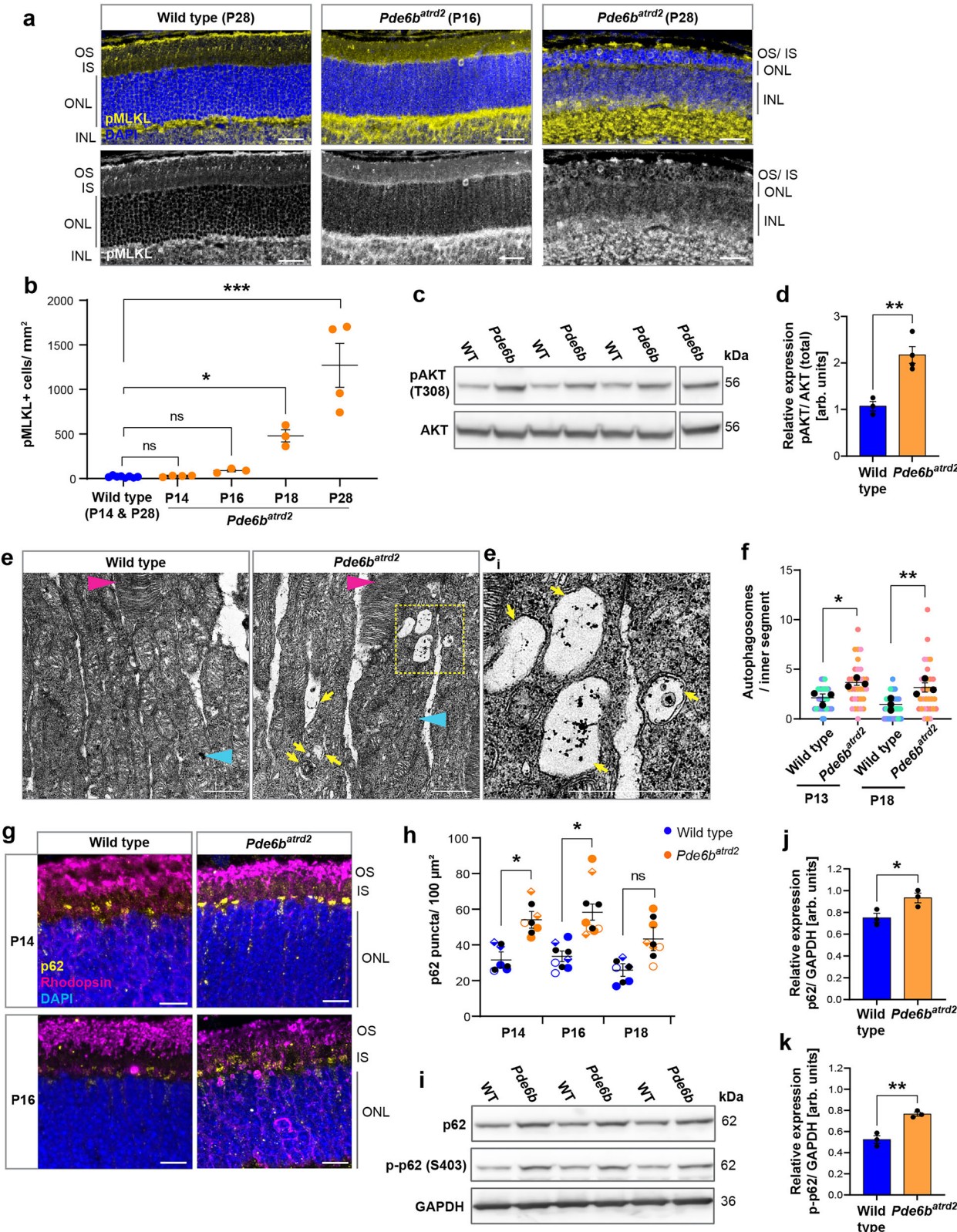

disruption of the dynamic process of autophagy, termed autophagic flux. Further work is needed to define the state of autophagy flux in RP.

Increased levels of mitochondrial ROS have recently been observed prior to photoreceptor loss in several mouse models of retinal degeneration, including an *Rpgr* knockout[52]. While mitochondrial dysfunction is not required to initiate necroptosis[53], mitochondrial ROS have been shown to promote autophosphorylation of

receptor interacting protein kinase 1 (RIPK1) and can therefore facilitate formation of the RIPK1/RIPK3/pMLKL necrosome complex[54,55]. RIPK3 can subsequently activate the mitochondrial pyruvate dehydrogenase complex, leading to enhanced aerobic respiration and increased generation of ROS[56,57]. This acts as a feedforward mechanism, further increasing necroptosis. Mitochondrial stress in *Rpgr* mutants could therefore facilitate photoreceptor death by

**Fig. 7 | Necroptosis and autophagy dysregulation are important cell death mechanisms in the *Pde6b*<sup>atrd2</sup> mouse model of RP. a** Increased numbers of pMLKL positive photoreceptors were observed in *Pde6b*<sup>atrd2</sup> mutants (scale bar = 25 µm). **b** Quantification of pMLKL positive photoreceptors per mm² in each retinal section. Wild type combines counts from P14 and P28 retinas from different mice (*n* = 8 animals in total). All other data points are from individual mice (P14 and P28: *n* = 4 animals per time point; P16 and P18: *n* = 3 animals per time point; bars show mean, error bars show SEM; multiple comparisons by 2-way ANOVA *$p$ = 0.027; ***$p$ = 0.0006). **c, d** Increased level of active pAKT relative to total AKT in retina lysates at P16 (*n* = 3 wild type and *n* = 4 mutant animals; bars indicate mean; error bars show SEM; **$p$ = 0.004 by *t*-test). **e** Transmission electron microscopy images of wild-type and mutant photoreceptors at P13 showing accumulating autophagosomes in mutants (scale bars = 500 nm; magenta arrowhead indicates outer segment; cyan arrowhead indicates inner segment; yellow arrows indicate

autophagosomes; region highlighted in yellow box is enlarged in **e**₁). **f** Quantification of autophagosomes per inner segment at P13 and P18 (colours indicate counts from individual mice *n* = 3 animals for each genotype at each time point, means for each animal overlaid in black; error bars indicate SEM; for each time point, means for individual mutant and wild type animals were compared by unpaired *t*-tests *$p$ = 0.034, **$p$ = 0.007). **g** Accumulation of p62 in *Pde6b*<sup>atrd2</sup> photoreceptor inner segments at P14 and P16 (scale bar = 10 µm). **h** Quantification of p62 puncta in inner segment. (Symbols indicate counts from individual mice; *n* = 3 animals per genotype at each time point; error bars show SEM; for each time point, means for individual mutant and wild type animals were compared by unpaired *t*-tests *$p$ = 0.026 (P14), $p$ = 0.015 (P16)). **i– k** p62 and p-p62 (S403) levels are increased in mutant retina lysates at P16 (bars indicate the mean; *n* = 3 mice per genotype; error bars show SEM; *$p$ = 0.04, **$P$ = 0.009 by unpaired *t*-test). Source data are provided as a Source Data file.

necroptosis. Indeed, inhibition of the mitochondrial stress response was shown to alleviate necroptosis in rat retinal neurons following ischemia-reperfusion injury[33]. The autophagy machinery can also form a scaffold for the necrosome, mediated by p62 recruitment of RIPK1, and this interaction promotes cell death by necroptosis rather than apoptosis[58]. It is therefore possible that the need to remove damaged mitochondria in *Rpgr* mutant photoreceptors leads to enhanced mitophagy, which in turn promotes necroptosis. Interestingly, an increased association of the autophagy regulators BECN1 and ATG5 with mitochondria has been shown in photoreceptor inner segments in a *Rho*<sup>P23H</sup> rat model of RP, suggesting increased mitophagy. Necroptosis was also identified as the primary mechanism of photoreceptor death in this model[20].

Necroptosis has a role in other neurodegenerative diseases, where autophagy dysregulation contributes to the pathology[59–63], and inhibition of necroptosis can reduce neural degeneration in animal models[59,61,63–66] and xenografted human neurons[67]. A greater understanding of the interplay between necroptosis, autophagy/mitophagy and mitochondrial stress in RP, therefore, will allow identification of potential therapeutic targets upstream of necroptosis pathway activation.

## Methods
### Animals
All experiments followed international, national and institutional guidelines for the care and use of animals. Animal experiments were carried out under UK Home Office Project Licenses PPL P1914806F and PP5301383 in facilities at the University of Edinburgh (PEL 719 60/2605) and were approved by the University of Edinburgh animal welfare and ethical review body. Mice were housed with a standard light/dark cycle at ambient temperature and humidity.

*Rpgr*<sup>Ex3d8</sup> and *Rpgr*<sup>ORFd5</sup> mice were generated using CRISPR/Cas9 genome editing in C57Bl-6J background as described in Megaw et al. 2024. Male hemizygous mutants and wild type littermates were used for all experiments. *Pde6b*<sup>atrd2</sup> mice were generated by N-ethyl-N-nitrosourea (ENU) induced mutation in sighted-C3H background strain as described[37]. A mix of male and female homozygous mutants and wild type littermates were used in each experiment.

### Single-cell RNA sequencing and analysis
Mice were sacrificed, eyes enucleated and placed into PBS. Retinas from one mutant male and one wild-type male littermate control at 18 months of age were used for each experiment. Retinas were dissected in NeuroCult Tissue Collection Solution (Stem Cell Technologies) and dissociated into single cells using the NeuroCult Enzymatic Dissociation Kit for Adult CNS Tissue (Stem Cell Technologies cat. 05715) according to the manufacturer's instructions. Cells were passed through a 40 µm FLOWMI cell strainer (SP Bel-Art) and live (DAPI-negative), single cells sorted using FACS Aria II. Barcoding, RNA

extraction and library preparation was carried out using Chromium Single Cell 3′ Reagent kits v3 (10X Genomics) according to the manufacturer's instructions. Libraries were sequenced using an Illumina NextSeq550 with a mid-output Flowcell, 150 cycles (130 M pair end reads, 2x 75 bp). The read alignment was performed with Kallisto[68] using locally built index and transcript-to-gene mapping based on the GRCm38 (release 98) gene and genome assembly files from Ensembl. The unfiltered count matrices produced by Kallisto were converted to SingleCellExperiment (SCE) objects and empty droplets were removed with EmptyDrops[69] using the default parameters. After removing all genes with counts of zero for all barcodes, the remaining genes were annotated with relevant features from Ensembl. Cell-level QC was performed with Scater[70], excluding cells with high amount of reads coming from mitochondrial genes (nmads > 3) (Supplementary Fig. 9). Using Seurat[71] we further filtered the count matrices to contain only protein-coding genes with non-zero counts in at least 10 cells and cells expressing at least 500 protein-coding genes, after which we merged the wild type and mutant libraries for each experiment, resulting in 2042 wild type and 2153 mutant cells for *Rpgr*<sup>ORFd5</sup> and 2299 wild type and 3277 mutant cells for *Rpgr*<sup>Ex3d8</sup> (Supplementary Table 2). Clustering of the cells for the two experiments was done with Seurat, by performing log-normalisation, centring and scaling, clustering based on the identified variable features using shared nearest neighbour graph and the Louvain algorithm with multilevel refinement with resolution = 0.5 to produce the UMAP plots (Supplementary Figs. 2a and 3a). Cluster annotation was initiated by identification of each cluster markers against the remaining clusters using Seurat's FindMarkers function (with min.pct = 0.25, logfc.threshold = 0.25) and selecting only upregulated genes with p_val_adj <0.05, followed by manual curation based on existing literature. The trajectory plots presented in Fig. 1a were generated using Slingshot[72]. Gene ontology analysis was performed using Enrichr[73–75].

### Immunohistochemistry
Mice were sacrificed, eyes enucleated and placed into Davidson's fixative (28.5% ethanol, 2.2% neutral buffered formalin, 11% glacial acetic acid) for 1 hr (cryosectioning) or overnight (wax embedding). For cryosectioning, eyes were removed from Davidson's fixative and placed into 20% sucrose in PBS buffer overnight for cryopreservation. Eyes were then embedded using optimal cutting temperature and kept at −80 °C until sectioned. For wax preservation, eyes were fixed in Davidson's fixative overnight at 4 °C. Following fixation, eyes were incubated successively in 70% v/v, 80% v/v, 90% v/v and 100% v/v ethanol, twice in xylene and then paraffin, each for 45 min per stage, using a vacuum infiltration processor. After sectioning, paraffin was removed by washing with 100% xylene and sections were rehydrated through an ethanol series (100%, 80%, 50%, 30%, 5 min each). For antigen retrieval, sections were boiled in sodium citrate pH6.4 for 2 × 10 min, then cooled to room temperature and washed in dH₂O. All

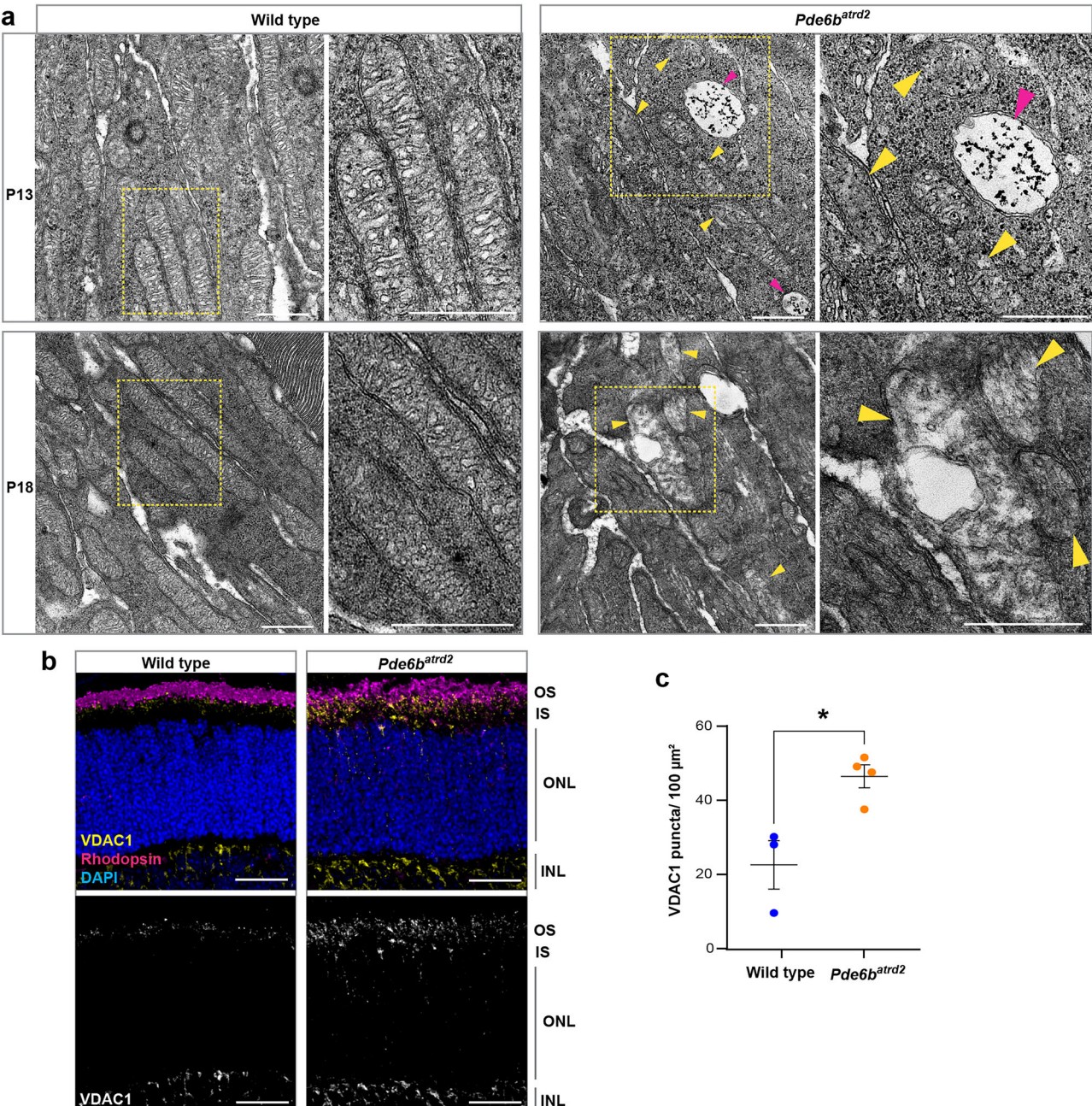

**Fig. 8 | Mitochondrial stress develops in *Pde6b^atrd2* mutant photoreceptors.**
**a** Transmission electron microscopy images of mutant and wild-type photo-receptor inner segments show mitochondrial swelling in *Pde6b^atrd2* mutant photo-receptors (yellow arrowheads) at P13 and P18. Large autophagosomes are also present in *Pde6b^atrd2* photoreceptor inner segments at P13 (magenta arrowheads)

(yellow boxed regions enlarged in right-hand panels; scale bars = 500 nm). **b** VDAC1 expression is increased in mutant photoreceptor inner segments at P14 compared to wild-type littermate controls (scale bar = 25 μm). **c** Quantification of VDAC1 puncta at P14 (*n* = 3 wild type animals and *n* = 4 mutant animals); error bars show SEM; *$p$ = 0.048 by unpaired *t*-test. Source data are provided as a Source Data file.

eye cups were uniformly aligned when embedded and, to control for eccentricity, all sections used for immunofluorescence included the optic nerve. Sections were post-fixed in acetone for 10 min at −20 °C, then blocked/permeabilised with 4% BSA (Sigma) and 0.2% Triton-X100 (Fisher) for 1 hr at RT and incubated with primary antibodies overnight at 4 °C. Sections were then washed in PBST (PBS + 0.2% Triton-X100), incubated with secondary antibodies for 2 hr at RT, washed in PBST, incubated in DAPI for 5 min at RT and mounted with coverslips using Prolong Gold (Invitrogen). Primary antibodies used were: mouse anti-Rhodopsin [4D2] (Abcam ab98887, 1:1000), rabbit anti-GFAP (Abcam ab7260, 1:500), rabbit anti-p62 (ENZO BML-

PW9860-0100, paraffin only, 1:300), rabbit anti-LAMP1 (Abcam ab24170, 1:500), rabbit anti-P2Y12 [E9J1J] (Cell Signaling Technology 69766, 1:200)rat anti-F4/80 [A3-1] (AbD Serotec MCA497EL, 1:500), rabbit anti-VDAC1 (Abcam ab15895, 1:200), rabbit anti-pMLKL [EPR9515(2)] (Abcam ab196436, 1:100), rabbit anti-cleaved caspase-3 (Cell Signaling Technology 9661, 1:400). Secondary antibodies used: Donkey anti-rabbit Alexa-488, Donkey anti-rabbit Alexa-594, Donkey anti-mouse Alexa-594, Donkey anti-rat Alexa-488 (all Invitrogen). Slides used for each experiment were stained on the same day, a master mix of antibodies with 1% BSA in PBST was made for each experiment and aliquoted between all slides. Confocal imaging was

performed on a Nikon A1+ Eclipse TiE inverted microscope or a Leica Stellaris inverted microscope. 40x dry and 60x oil immersion lenses were used. To further control for retinal eccentricity, images were taken from regions approximately half way between the optic nerve and the periphery in all samples. Data was acquired using NIS Elements AR software (Nikon Instruments Europe, Netherlands) or LASX software (Leica Biosystems). All images for a single experiment (or single time point for an experiment) were taken on the same day and identical imaging settings were used for all images in each experiment. Each image was obtained using the same number of optical projections per z-stack and with the same inter-projection distance. Z-stacks were processed and analysed in Fiji (ImageJ).

For quantifying accumulation of p62 and LAMP1 in *Rpgr^Ex3d8* photoreceptor inner segments, sections were imaged from 3 individual mice, with at least 2 sections imaged from each animal. For quantification of p62 accumulation in *Pde6b^atrd2* photoreceptor inner segments, sections were imaged from 3 animals per genotype at each time point with 2 sections imaged from each animal. For VDAC1 quantification, single sections were imaged from 5 individual mice (*Rpgr^Ex3d8* mutant and wild type control at 12 months, *Rpgr^Ex3d8* at 18 months), 4 individual mice (*Pde6b^atrd2* at P14) or 3 individual mice (wild type controls at 18 months and P14). To quantify pMLKL positive cells in *Rpgr^ORFd5*, *Rpgr^Ex3d8* and *Pde6b^atrd2*, whole retina sections were imaged at 40x using the LASX Navigator function. For *Rpgr^ORFd5*, *Rpgr^Ex3d8* and respective wild type controls, we imaged at least 3 retina sections from 3 animals for each genotype at each time point. For *Pde6b^atrd2* one retina section was imaged from each animal: retinas from 4 *Pde6b^atrd2* mice were imaged at P14 and P28, retinas from 3 *Pde6b^atrd2* mice were imaged from P16 and P18, retinas from 8 wild type mice were imaged for controls (mix of P14 and P28).

### Protein extraction and western blotting
Retinas were lysed in 50 mM Tris pH8.0, 150 mM NaCl, 1% NP-40 buffer containing protease inhibitors and phosphatase inhibitors. Protein samples were separated by SDS-PAGE and electroblotted onto nitrocellulose membranes (Biorad) using the Trans-Blot Turbo System for 10 min (Biorad). Non-specific binding sites were blocked by incubation of the membrane with 5% non-fat milk in TBS containing 0.1% Tween 20 (TBST) for 1 hr. Proteins were detected using primary antibodies diluted in blocking solution (4% Bovine Serum Albumin in TBST). Primary antibodies used were: rabbit anti-AKT [C67E7] #4691, rabbit anti-pAKT [D25E6] #13038, rabbit anti-caspase 8 [D35G2] #4790, rabbit anti-cleaved-caspase 8 [D5B2] #8592, rabbit anti-NDUFS1 [E4K3E] #70264, rabbit anti-mtCO1 [E2I2R] #55159, rabbit anti-RISP [95231], rabbit anti-CYTC [D18C7] #11940, rabbit anti-pSQSTM1 (p-p62) S403 [D8D6T] #39786 (all Cell Signalling Technology, 1:1000), mouse anti-p62 [2C11] (Abcam ab56416, 1:1000), rabbit anti-aTUB (Abcam ab4074, 1:2000), mouse anti-GAPDH [6C5] (Sigma MAB374, 1:2000). Following washing in PBST, blots were incubated with the appropriate secondary antibodies conjugated to horseradish peroxidase (Cell Signalling Technology, 1:10,000) and chemiluminescence detection of Super Signal West Pico detection reagent (Pierce) by high resolution image capture using the ImageQuant LAS4000 camera system (GE Healthcare). Images were transferred to Fiji/ImageJ and mean pixel intensity of protein bands measured for quantification, with an equal area of blot assessed across all bands.

### Transmission electron microscopy
3 mutant and 3 wild type mice were imaged for each genotype at each time point. *Rpgr^Ex3d8* mice and wild type littermates were sacrificed at 6, 12 and 18 months old, *Pde6b^atrd2* mice and wild type littermates were sacrificed at P13 and P18. Mice were euthanised by transcardial perfusion using fixative (0.2 M sodium cacodylate pH 7.4, 5% glutaraldehyde, 4% paraformaldehyde). Eyecups were enucleated and placed in 1 mL of fixative. After 30 min, the cornea and lens were removed,

then left to incubate in fixative for a total of 2 hr at room temperature. Thereafter, retinas were either:

1. Washed in 0.1 M phosphate buffer (pH 7.4), post-fixed with 1% osmium tetroxide (Electron Microscopy Science) and dehydrated in an ethanol series prior to embedding in Medium Epoxy Resin (TAAB). Ultrathin (75 nm) sections of the retina were then stained with aqueous uranyl-acetate and lead citrate and then examined with a Hitachi 7000 electron microscope (Electron Microscope research services, Newcastle University Medical School).
2. Polymerised in 4% agarose (Genemate E-3126-25). 150 μm sections were collected into MilliQ water using a vibratome. Sections were then stained in 1% osmium tetroxide 0.1 M sodium cacodylate for 40 min, with rocking at room temperature, covered. After rinsing in MilliQ water, the sections were stained with 1% uranyl acetate in 0.2 M maleate buffer, pH 6.0, for 1 hr, with rocking at room temperature, covered. The sections were rinsed in MilliQ water and dehydrated in a series of ethanol washes (50%, 70%, 90%, 100%, 100%) for 15 min each, followed by two 100% acetone washes, 15 min each. The sections were then embedded in Epon-12 resin by sandwiching the sections between two sheets of ACLAR (EMS 50425-10) and leaving them at 60 °C for 48 hr. Ultrathin silver sections (60 nm) were placed on copper slot grids (EMS FF2010-CU) and post-stained in 1.2% uranyl acetate in MilliQ water for 6 min, followed by staining in Sato's lead (a solution of 1% lead acetate, 1% lead nitrate, and 1% lead citrate; all from Electron Microscopy Sciences) for 2 min. Sections were imaged on a JEOL JEM-1400 electron microscope.

### Mass spectrometry
Dark-adapted mice were maintained in constant darkness for 12 hr overnight prior to retina harvesting (also performed in the dark with infrared illumination). Retinas were lysed in 2% SDS and kept at −80 °C until testing. 2 retinas from 1 mouse were used per biological replicate, with 5 biological replicates per group (mutant and wild type). In all cases, cell lysate samples were prepared and analysed as described[23].

### Mitochondrial stress test assay
Xfe24 Seahorse cartridges (Agilent) were hydrated overnight prior to the experiments in Seahorse XF Calibrant (Agilent) at 37 °C in a non-CO$_2$ incubator. Mesh inserts from Xfe24 Islet plates were pre-coated in Cell-Tak (Corning) using the sodium bicarbonate adsorption method. On the day of the experiment, mice were sacrificed by dislocation of the neck, eyes were enucleated and the retinas dissected in PBS. 4-6 punches were cut from each retina using a 1 mm biopsy punch, transferred to pre-coated mesh inserts (ganglion cell side down) and placed in individual wells of Xfe24 Islet plates with 500 μl of Seahorse DMEM media pH 7.4 (Agilent) supplemented with 10 mM glucose, 2 mM pyruvate and 2 mM Glutamine. Biopsies were incubated at 37 °C in a non-CO$_2$ incubator for 20 min prior to running the assay. 8-10 biopsies from 1 mouse were used as technical replicates for each experimental sample, with a total of 4 biological replicates (4 mutant and 4 wild type mice). Mitochondrial oxygen consumption rates were measured using an Agilent Seahorse Xfe/XF Analyser using the standard protocol (Mix, 2 min; Wait, 2 min; Measure, 3 min). After three basal OCR measurements were obtained, OCR was measured following sequential injection of mitochondrial drugs (3 measurements per drug injection); 2 μM oligomycin (port A), 1 μM FCCP (port B), 0.5 μM antimycin A + 0.5 μM rotenone (port C).

### Image analysis
Image analysis was performed using Fiji/ ImageJ. For punctate staining (p62, LAMP1, VDAC1) the inner segment area was defined manually for each image as a region of interest (ROI) and the number of puncta was calculated from maximum intensity projections of the relevant channel. Images were filtered to remove noise using 'Despeckle' and the

threshold adjusted to create a binary image, with the level set so that individual puncta could be identified. The same threshold settings were used for all images for each staining. Puncta within the defined ROI were counted using 'Analyse Particles' and counts were normalised to the total area of the ROI. To quantify pMLKL positive cells, whole retina sections were imaged at 40x using the LASX Navigator function and cells were counted manually from maximum intensity projections of 6 μm z-stacks. Cell counts were normalised to the total area of the ONL.

## Statistics and reproducibility

All statistical analysis was carried out using GraphPad Prism 9 (version 9.5.1; GraphPad software, USA) as described in the text. Sample sizes were chosen to be large enough to allow statistical significance to be determined while keeping the number of animals used to a minimum, in accordance with UK NC3Rs principles. However, no statistical method was used to predetermine sample size. The experiments were not randomised and the investigators were not blinded to allocation during experiments. However, all image acquisition and analysis was blinded to genotype where possible by assigning randomised numbers to each image with a code that was not known until all analysis was completed. No data were excluded from the analyses. To determine statistical significance, unpaired $t$-tests were used to compare between two groups, with Welch's correction for unequal variance. Where more than one retinal section or sample from the same animal was analysed, all statistical analysis was based on the mean values for each animal. Multiple comparisons across time points were analysed by 2-way ANOVA. The mean ± the standard error of the mean (SEM) is reported in the corresponding figures as indicated. Statistical significance was set at $P < 0.05$.

## Reporting summary

Further information on research design is available in the Nature Portfolio Reporting Summary linked to this article.

## Data availability

Single cell RNAseq data generated in this study are available in the Gene Expression Omnibus database under accession code GSE294049. Source data are provided with this paper.

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

## Acknowledgements

This work was funded by the Wellcome Trust (RM; 219607/Z/19/Z), Fight for Sight (FN; 5179 / 5180) and the Medical Research Council (P.M., L.M., L.N., M.H.; P.M.: MC_UU_00007_14, MR_Y015002_1).

## Author contributions

R.M. and P.M. conceived the project. R.M., F.N., L.N., M.H., and P.M. designed the experiments. F.N., L.M., L.N., and M.H. performed the experiments. F.N., L.M., L.N., M.H., P.M., and R.M. analysed the data. P.M. and R.M. coordinated the study and provided guidance. F.N., P.M., and R.M. wrote the paper. All of the authors discussed the results and approved the final version of the manuscript.

## Competing interests

The authors declare no competing interests.
