## [Transparent Peer Review file · Nature Communications]

Autophagy disruption and mitochondrial stress precede photoreceptor necroptosis in multiple mouse models of inherited retinal disorders

Corresponding Author: Dr Roly Megaw

Version 0:

Reviewer comments:

Reviewer #1

(Remarks to the Author)

In this study the authors have evaluated whether there were common pathways associated with photoreceptor death in two models of inherited retinal degeneration. They use scRNAseq on dying photoreceptors from two animal models to identify common anomalies in autophagy and mitochondrial dysfunction. Overall, the manuscript is well presented and written and the scRNAseq data interesting. The combination of histology, molecular biology, and metabolic analysis is a strength of this study. However, there are some significant errors in the presentation of the retinal figures (incorrect labelling of layers), that make me very concerned about the robustness of the data overall. In addition, some information is missing from the methods.

Below I outline suggested changes and concerns/queries:

1. Methods: Some aspects of the methods need more detail: For all immunocytochemistry and quantification, was retinal eccentricity controlled for? Retinal degeneration in many models is patchy and I note the variation in photoreceptor density across eccentricity has been previously reported in these models (Megaw et al 2024). Ensuring that similar eccentricities are used in all analyses is important.
 - a. Was there masking and randomization performed for all experiments?
 - b. What were the wildtype and background strains of your mice? Given this is an X linked mutation what were the sexes of your mice. This should be stated in the methods, even if you have provided a reference to another paper.
 - c. Animal numbers: Its not clear to me how many animals were used in the scRNAseq and how many animals were used in your immunocytochemistry and EM experiments. Please include these details in the methods.
 - d. Image analysis: please include more information on how you quantified p62, LAMP1 etc. Were multiple sections per retinae evaluated, across eccentricity. Please see my concerns raised about this data below.
2. Statistical analysis. You mentioned in the methods that statistical analysis is provided in the text where there are more than two groups being compared. However, I couldn't find this information for Figure 2d, Figure 3, Figure 5 or Figure 6. I assume for all these comparisons that there was correction for the multiple measurements in the same animal - ie a repeated measures two way ANOVA or some other mixed linear model? Please provide this information – perhaps in a separate section in the methods that includes all statistical analysis would be helpful.
3. Figure 1/Results: You state that the two strains of RpgR mice either manifest cone loss or rod loss initially, yet you only evaluate rod death in your scRNAseq at 18 months. Are you able to provide data that confirms the level of rod vs cone loss in your two strains at 18 months? For example, a vertical section/quantification of a cone marker would be helpful (maybe arrestin 3, or PNA). It would be interesting if you were able to identify whether similar pathways contribute to cone loss? Also, if the rod vs cone loss differs across the two strains, could this impact the analyses that you have performed?
4. Supplementary Figure 1: you state that gliosis is present prior to photoreceptor loss in “both lines” (see line 73), however I can only see data for RpgREx3d8 in the supplementary figure. Please provide data from the RpgRORFd5 or change your sentence. Also – to be more compelling, it would be useful to quantify the extent of gliosis in the two mouse strains relative to the loss in rods and cones. To do this, the number of Muller cells labelled for GFAP could be quantified and correlated with

cone or rod density (maybe ONL thickness as a gross measure of photoreceptor degeneration).

5. Supplementary Figure 2 & 3: please correct the labels on your retinal sections (panel e) – IPL and OPL are incorrect. Also, although TMEM119 is a good marker of microglia in the brain, it doesn't label retinal microglia particularly well. I note that your TMEM119 labeling is very faint (almost unrecognizable). To be more convincing, I would recommend repeating these immunocytochemistry experiments using more conventional markers – perhaps even those that differentiate between inflammatory monocytes (eg CCR2+) and microglia – eg P2Y12R, or Iba1.

6. Figure 3, 5 and 7: Quantification of VDAC1, P62 and LAMP1 in retina sections. I am not convinced that quantification of these proteins using immunofluorescence is a robust method for quantifying protein levels in a retina. The problem is that retinal section thickness can vary by up to 20% and careful control of the immunocytochemistry processing (ie same antibody aliquot, same imaging settings, performing the IMCC on the same slides etc) is necessary. None of these important controls have been described here. Also considering the potential for bias with these analyses was any form of masking applied? More information is required to be satisfied that these differences are real. Given that P62 and LAMP1 show a punctate style of immunoreactivity, a more robust way of quantifying p62 and LAMP1 might be to quantify the “puncta” per ONL area. Or alternatively use some other method of quantification to support these observations.

Figure 6 and 7: quantification of pMLKL+ cells. Please use a similar method to document results for these two experiments. Here pMLKL+ cells per section (Figure 6D) or per μm^2 (Figure 7B) have been quantified. On the one hand, we do not know how large a “section” is – my advice is to convert Figure 6D to μm . And on the other, in Figure 7b the numbers are crazily small – perhaps convert to something more meaningful (eg 100 μm , mm?).

Reviewer #2

(Remarks to the Author)

This manuscript by Newton et al., describes the molecular and cellular phenotype of two novel IRDs mouse models previously generated by the authors, namely Rpgr mouse models (doi: 10.1038/s41467-024-48639-w). By using scRNAseq approach and cellular biology techniques they show that mitochondrial morphology and function, as well autophagy, are impaired in RpgrEx3d8 (manifesting cone disease-features) and RpgrORF5 strain (manifesting rod disease-features). Moreover, they found that necroptosis, rather than apoptosis, represents the main cell death pathway triggering photoreceptor cell death. Overall, the accurate molecular and cellular phenotyping performed by the authors support the occurrence of both autophagy and mitochondrial dysfunction in necroptotic retina of IRDs mouse models, which is noteworthy in the field. However, the mechanistic details underlying how RPGR functions to regulate mitochondrial function and autophagy (and thus trigger necroptosis -as postulated in the Discussion and Supplementary Figure 7-) are still missing.

Inherited retinal dystrophies (IRDs) are a heterogeneous group of inherited disorders with over 280 identified genetic mutations. Given the limited therapies available for these disorders, the “philosophical” aim of this work is highly relevant. Indeed, the IRDs high genetic heterogeneity complicates the development of “low-cost” mutation-independent therapeutical approaches that may potentially benefit a broad group of patients with the same condition regardless of their genotype. Newton et al. consider that RpgrEx3d8 and RpgrORF5 may provide models with which to identify shared pathway changes across the IRD spectrum. The latter is essential to develop novel therapeutic approaches targeting shared pathways and thus applicable to a broad group of IRDs of patients. Therefore, although the overall relevance of this study (accurate phenotyping of two novel IRD mouse models), the novelty of the insights provided might be overstated by the authors, and I am referring mainly to the impact on IRD therapeutical perspectives. In recent years multiple findings have shed light on the important role of autophagy in the maintenance of photoreceptor homeostasis, and many papers described autophagy and mitochondrial damage as hallmark of photoreceptors neurodegeneration in IRDs (10.3389/fphys.2018.01008; <https://doi.org/10.1038/s41419-018-0855-8>; 10.1016/j.preteyeres.2016.08.001; <https://doi.org/10.1016/j.mam.2021.101038>; <https://doi.org/10.1016/j.nbd.2021.105405>).

Similarly, although it has been a longstanding belief that apoptosis is the general pathway for photoreceptor death in retinal degeneration, its role in photoreceptor degeneration, especially in rods, has been questioned over the last two decades (DOI: 10.2174/156652412800620048). Other non-apoptotic mechanisms, including necroptosis, have been proposed as the dominant mode of cell death in RP (doi: 10.1038/s41419-018-0855-8), by several authors (<https://doi.org/10.1016/j.nbd.2024.106463>; <https://doi.org/10.1038/cdd.2013.191>; <https://doi.org/10.1016/j.exer.2020.107922>).

Overall, the work is well performed and provides convincing data on the coexistence of mitochondrial dysfunction-autophagy alterations-necroptosis activation in IRDs photoreceptors. Nevertheless, I have several comments, divided into Major, Minor and Format/Stylistic:

Major

- The evaluation of two transgenic strains harbouring mutations in the same gene is rather a poor approach to identify common altered pathway to be therapeutically targeted in a pathology, i.e. inherited retinal diseases, for which thousands of variants and hundreds of genes have been identified. At the end of the manuscript the authors briefly analyse -only at cellular level- the Pde6Batrd2 mouse model and describe some phenotypic traits also observed in Rpgr strains. However, as stated before, alterations in autophagy, cell death and mitochondrial damage are already well known common hallmark of

IRDs, therefore the author should provide a better molecular characterization of Pde6Batrd2 to strengthen their conclusions and the novelty of the overall work.

- The authors postulated that ROS may promote the activation of necroptosis and trigger a “ROS induced by ROS” cycle which further increments cell death (lines 360-366). I suggest a careful evaluation of ROS levels in the three mouse models included in the manuscript – e.g. by using ex-vivo retinal explants- to strengthen this hypothesis.

- Many details concerning the assessment of the mitochondrial physiology are lacking, such as the expression pattern of OXPHOS proteins, the enzymatic activity of the respiratory complex, ATP production and membrane potential.

- All the mechanistic hypothesis presented to explain the molecular basis giving rise to the cellular phenotype are based on the scRNAseq, whereas the proteins encoded by the differentially expressed genes are poorly characterized (e.g. total levels, changes in post-translational modifications etc). That is pivotal, due to the amount of post-transductional modifications regulating the pathways that the authors propose to be involved in IRD pathogenesis. Moreover, at technical level, western blots may provide more reliable and quantifiable data in comparison to immunohistochemistry.

Minor

- In the Figure 2 a marked increase in mitophagy is clearly detectable: a great amount of autophagosomes are engulfing mitochondria. This aspect is very interesting and should be further studied (e.g. by performing quantifications here and in Figure 4 as well, by checking possible alterations in the expression of genes specifically involved in mitophagy).

- Alterations in general autophagy and mitophagy are frequently associated with alterations in mitochondrial biogenesis which is used to compensate the loss of dysfunctional mitochondria and repopulate the mitochondrial pool with newborn organelles. It would be interesting to check if master genes/proteins regulating mitochondrial biogenesis are altered in these models and go further deep in their molecular characterization.

- In lines 89-92 authors observed that “Immunofluorescent staining of retina cryosections confirmed cells, positive for the pan-macrophage marker F4/80 and the microglial marker TMEM119, invading outer layers of RpgrORFd5 and RpgrEx3d8 retinas at 18 months, something not observed in wild type at this age (Supplementary Figs. 2e, 3e)”. I cannot see any F4/80 or TMEM119 positive cell in the Supplementary Fig. 3e and that would fit with the scRNAseq data (line 95-97). Therefore, the authors should discuss about this difference between the two strains, given that inflammation is a phenotype very linked to neurodegeneration.

- On the same page of some mayor comment, I miss western blot analysis in Figure 5 not only for VDAC, that is a structural protein, but also for proteins involved in mitochondrial function (such as OXPHOS proteins). The latter will significantly strengthen the message.

Format/Stylistic

- I miss an introductive section on Rpgr and the encoded proteins (acronym, function, presence of retina-specific isoforms etc.) and, eventually, a schematic representation of the mutant proteins.

- The age of the animals is not always indicated in the figure legend (e.g. Figure 3).

- Check that all the figures in the main body of the text are called out in sequential order (e.g. “(Fig.2c)”, line 194 or “Supplementary Fig.4c”, line 257)

- Complete the information regarding the “n” used for statistics, by specifying the number of sections and/or retinas and/or animals. This should usually be included in each figure legend.

Version 1:

Reviewer comments:

Reviewer #1

(Remarks to the Author)

I thank the authors for their careful consideration of this reviewers suggestions. All my concerns and comments have been addressed. I think this study will be a terrific addition to the corpus of knowledge on retinal degenerations. Congratulations!

Reviewer #2

(Remarks to the Author)

The authors have been quite responsive to my comments and they have addressed most of my concerns. I think that their initial conclusions have been properly strengthened in this new version of the manuscript.

MRC Human
Genetics
Unit

THE UNIVERSITY
of EDINBURGH

Nature communications reviewers

14th February 2025

Dear reviewers,

RE: Submission NCOMMS-24-46257

Many thanks for taking the time to read our article, "*Autophagy disruption and mitochondrial stress precede photoreceptor necroptosis in multiple mouse models of inherited retinal disorders*" and providing such useful feedback.

We have revised our manuscript which, alongside this 'Response to Reviewers' document, incorporates all changes suggested by yourselves and highlight the changes we have made. We will address each of the comments in turn and all changes in the manuscript text file are highlighted with track changes.

We are satisfied we have addressed all of your comments. This has lengthened our paper and we believe the new data are significant contributions to the field. We hope that you will be satisfied with the changes we have made to the manuscript.

With best wishes

Yours sincerely

Dr. Roly Megaw

Clinician Scientist, MRC HGU

Medical Research Council Human Genetics Unit at the University of Edinburgh

Institute of Genetics and Cancer, Western General Hospital, Crewe Road, Edinburgh EH4 2XU

T: +44 (0)131 651 8500 ed.ac.uk/mrc-human-genetics-unit @MRC_HGU

We thank the reviewers' consensus that our '*combination of histology, molecular biology, and metabolic analysis is a strength of this study*' (reviewer 1) which was '*well performed and provides convincing data on the coexistence of mitochondrial dysfunction-autophagy alterations-necroptosis activation in IRDs photoreceptors*' (Reviewer 2). We acknowledge the lack of detail in some sections of the methods in the original submission, which we have now rectified, along with errors in labelling caught by our reviewers. We are grateful for their thoughtful comments, careful reading and constructive suggestions to improve our revised resubmission.

Reviewer #1:

(#1) Methods: Some aspects of the methods need more detail: For all immunocytochemistry and quantification, was retinal eccentricity controlled for? Retinal degeneration in many models is patchy and I note the variation in photoreceptor density across eccentricity has been previously reported in these models (Megaw et al 2024). Ensuring that similar eccentricities are used in all analyses is important.

- a. Was there masking and randomization performed for all experiments?
- b. What were the wildtype and background strains of your mice? Given this is an X linked mutation what were the sexes of your mice. This should be stated in the methods, even if you have provided a reference to another paper.
- c. Animal numbers: Its not clear to me how many animals were used in the scRNAseq and how many animals were used in your immunocytochemistry and EM experiments. Please include these details in the methods.
- d. Image analysis: please include more information on how you quantified p62, LAMP1 etc. Were multiple sections per retinae evaluated, across eccentricity. Please see my concerns raised about this data below.

RESPONSE #1 We stress that, for all immunocytochemistry and quantification, retinal eccentricity was controlled for. We have edited the methods section to inform the reader, with lines 740 – 741 now reading 'All eye cups were uniformly aligned when embedded and, to control for eccentricity, all sections used for immunofluorescence included the optic nerve.', and lines 757 – 759 now reading 'To further control for retinal eccentricity, images were taken from regions approximately half way between the optic nerve and the periphery in all samples.'

- a. Masking was performed in all experiments. Genotypes of samples were concealed during image acquisition and images were assigned randomised numbers with a code that was not known until after analysis was completed. Our experiments required specific mice genotypes to be used (with wild-type littermates used as controls). As such, randomisation was not possible. To reduce bias, all suitable mice, once genotypes were known, were used in experiments. In addition to preventing bias, this also reduced animal use, in keeping with NC3Rs policy. We hope this reassures the reviewer that this achieved the same objective; namely, that there was no bias in animal selection.
- b. All mice used in these studies, including wild-type controls were on a C57Bl6/J background. Given that the gene is X-linked, all experiments performed using our *Rpgr* mutant strains used only male mice (including wild-type littermate controls). However, experiments performed using the *Pde6b* mutant strains used both male and female mice, in order to avoid sex bias and improve the generalisability of our research findings. We feel this especially important, given our long-term goal of identifying therapeutic targets. We have amended our methods section to inform the

Medical Research Council Human Genetics Unit at the University of Edinburgh

Institute of Genetics and Cancer, Western General Hospital, Crewe Road, Edinburgh EH4 2XU

T: +44 (0)131 651 8500 ed.ac.uk/mrc-human-genetics-unit MRC_HGU

2

reader, stating on lines 680 – 685 that ‘*Rpgr*^{Ex3d8} and *Rpgr*^{ORFd5} mice were generated using CRISPR/Cas9 genome editing in C57BI-6J background as described in Megaw et al. 2024. Male hemizygous mutants and wild-type littermates were used for all experiments. *Pde6b*^{atrd2} mice were generated by N-ethyl-N-nitrosourea (ENU) induced mutation in sighted-C3H background strain as described³⁷. A mix of male and female homozygous mutants and wild-type littermates were used in each experiment’.

- c. Each of our two scRNASeq experiments were performed on one mutant and one wild-type littermate control and this has been highlighted in our methods section (see lines 692 – 694). Each figure legend states the number of mice used in each experiment (including immunocytochemistry and EM experiments), but we have also amended the methods to inform the reader on animal numbers (see lines 765 – 777 and 801 – 803).
- d. We have amended the methods section to include more information on how we quantified p62, LAMP1 and our other immunohistochemistry experiments. To answer the reviewer’s specific comments, eccentricity was controlled for (see above) and multiple sections per retinae were evaluated for each animal. This is reflected in our graphs, where measurements of individual retinal sections for each animal are represented by the same symbol.

(#2) Statistical analysis. You mentioned in the methods that statistical analysis is provided in the text where there are more than two groups being compared. However, I couldn’t find this information for Figure 2d, Figure 3, Figure 5 or Figure 6. I assume for all these comparisons that there was correction for the multiple measurements in the same animal - ie a repeated measures two way ANOVA or some other mixed linear model? Please provide this information – perhaps in a separate section in the methods that includes all statistical analysis would be helpful.

RESPONSE #2 We have added a separate section in the methods to include details of all statistical analyses used. These appear on lines 869 – 875 of the new manuscript. To answer Reviewer 1’s specific comments, we have used ANOVA to assess statistical significance where comparisons were made across multiple time points (Figure 6d, Figure 7b). We have not compared time points in Figure 2d, Figure 3, Figure 5c and Figure 7e since animals at different ages were stained and imaged on different days. Comparisons between mutant and wild type were assessed by unpaired t tests. Where multiple measurements were taken from the same animal, t tests were performed using the mean value for each animal. Where relevant, we have added the mean values for each animal to our graphs to clarify this.

(#3) Figure 1/Results: You state that the two strains of *Rpgr* mice either manifest cone loss or rod loss initially, yet you only evaluate rod death in your scRNAseq at 18 months. Are you able to provide data that confirms the level of rod vs cone loss in your two strains at 18 months? For example, a vertical section/quantification of a cone marker would be helpful (maybe arrestin 3, or PNA). It would be interesting if you were able to identify whether similar pathways contribute to cone loss? Also, if the rod vs cone loss differs across the two strains, could this impact the analyses that you have performed?

RESPONSE #3 The focus of this study was to better understand the pathway changes that lead to rod photoreceptor death in Retinitis Pigmentosa. As such, we concentrated on pathway changes in the rod populations of our scRNAseq experiments. The reviewer questions whether differing rod vs cone loss across the strains could impact our analyses.

Medical Research Council Human Genetics Unit at the University of Edinburgh

Institute of Genetics and Cancer, Western General Hospital, Crewe Road, Edinburgh EH4 2XU

T: +44 (0)131 651 8500 ed.ac.uk/mrc-human-genetics-unit MRC_HGU

Although rod death (and subsequent reduction in RDCVF secretion) is thought to impact on cone function and survival in RP, the reverse (cone loss) is not thought to affect rod viability. Further, the distinct advantage of scRNASeq over conventional bulk RNA sequencing is the pure populations of cell types that can be isolated according to characteristic gene expression (see supp fig 2 and 3 and supp table 1). We are thus confident that the cell intrinsic signatures detected in our mutant rods are cell autonomous and not impacted by other cell types. This is supported by our downstream analysis with immunohistochemistry, immunoblotting and electron microscopy studies.

The reviewer notes that, of the 2 *Rpgr* models we used, each ‘manifest cone loss or rod loss initially’. We stress that we do not claim this, but rather that they variably manifest rod versus cone ‘disease’, as evidenced by primary rod versus cone dysfunction on electroretinography (Megaw et al., 2024). The reviewer does, however, raise an interesting point and so we have quantified retinal cone populations in our *Rpgr*^{Ex3d8} mice using immunohistochemistry of whole retinal slices (see **‘Rebuttal Figure 1. Absence of cone loss in *Rpgr*^{ORFd5} and *Rpgr*^{Ex3d8} mice’**). Interestingly, despite the loss of cone function, there is no measurable cone degeneration, suggesting perturbation of *Rpgr* expression leads to compromised cone phototransduction prior to cell death. We have therefore rephrased line 70 – 76 so as to focus on rod degeneration in our models.

As also suggested by Reviewer 1, we have analysed the cone populations in our scRNASeq datasets for pathway changes. Though these isolated cone populations were, as expected, very small in number in each experiment, an upregulation in activity of mitochondrial biogenesis and mitochondrial function genes was seen, similar to that seen in rods. Similar, though less profound, upregulation of necroptotic gene activity was observed. We have included this as a rebuttal figure (see **‘Rebuttal Figure 2. Increased expression of genes required for mitochondrial biogenesis and function and necroptosis observed in *Rpgr* mutant cone photoreceptors’**) but have chosen not to include it in our revised manuscript as we feel it distracts from the main narrative of the work, namely defining rod dysfunction in RP. We hope the reviewer understands our reasoning for this.

(#4) Supplementary Figure 1: you state that gliosis is present prior to photoreceptor loss in “both lines” (see line 73), however I can only see data for *Rpgr*Ex3d8 in the supplementary figure. Please provide data from the *Rpgr*ORFd5 or change your sentence. Also – to be more compelling, it would be useful to quantify the extent of gliosis in the two mouse strains relative to the loss in rods and cones. To do this, the number of Muller cells labelled for GFAP could be quantified and correlated with cone or rod density (maybe ONL thickness as a gross measure of photoreceptor degeneration).

RESPONSE #4 We have now measured the reactive gliosis in our *Rpgr*^{ORFd5} line and have quantified levels in both our lines, doing so by correlating GFAP positive muller glia with ONL thickness. The analysis shows significant upregulation in both models, which we have added to Supplementary Figure 1.

(#5) Supplementary Figure 2 & 3: please correct the labels on your retinal sections (panel e) – IPL and OPL are incorrect. Also, although TMEM119 is a good marker of microglia in the brain, it doesn’t label retinal microglia particularly well. I note that your TMEM119 labeling is very faint (almost unrecognizable). To be more convincing, I would recommend repeating these immunocytochemistry experiments using more conventional markers – perhaps even those that differentiate between inflammatory monocytes (eg CCR2+) and microglia – eg P2Y12R, or Iba1.

RESPONSE #5 We have corrected the retinal section labels in Supplementary Figures 2 & 3. We have repeated the microglia staining, this time using the reviewer's recommended microglia marker, P2Y12. The staining identifies P2Y12^{High} F4/80^{Low} cells with amoeboid morphology present in the photoreceptor layer of *Rpgr*^{Ex3d8} and *Rpgr*^{ORFd5} retinas. It is likely that these are activated resident microglia. We also observe a second population of F4/80^{High} cells with much lower P2Y12 expression, located between the photoreceptor outer segments and RPE. Given this population express P2Y12, they could be another subset of microglia and we did not detect any P2Y12 negative cells in the sections analysed. However we cannot rule out a contribution from inflammatory monocytes. Crucially, neither of these cell populations were observed in retinal sections from age-matched wild-type mice, although inactive microglia (P2Y12^{High} cells with ramified morphology) are clearly present in the IPL. We present this staining in our updated Supplementary Figures 2e and 3e and have adjusted lines 97 – 107 in the manuscript accordingly.

(#6) Figure 3, 5 and 7: Quantification of VDAC1, P62 and LAMP1 in retina sections. I am not convinced that quantification of these proteins using immunofluorescence is a robust method for quantifying protein levels in a retina. The problem is that retinal section thickness can vary by up to 20% and careful control of the immunocytochemistry processing (ie same antibody aliquot, same imaging settings, performing the IMCC on the same slides etc) is necessary. None of these important controls have been described here. Also considering the potential for bias with these analyses was any form of masking applied? More information is required to be satisfied that these differences are real. Given that P62 and LAMP1 show a punctate style of immunoreactivity, a more robust way of quantifying p62 and LAMP1 might be to quantify the "puncta" per ONL area. Or alternatively use some other method of quantification to support these observations.

RESPONSE #6 We can reassure the reviewer that, for our immunofluorescence quantification experiments, the same antibody aliquot, the same slide type and the same imaging settings were used throughout all experiments. As mentioned above, retinal eccentricity was controlled for and, although 'retinal section thickness can vary by up to 20%', each image was obtained using the same number of optical projections per z stack and with the same inter-projection distance. Thus, the same thickness of retina was uniformly imaged. Masking was applied when analysing the data, which, following binning of the relevant area (inner segments, defined manually for each image), was performed using an automated macro on image j (sum intensity projection of z-stack, followed by measurement of standard deviation of pixel intensity within the defined inner segment region in the relevant channel). Thus, we have applied sound scientific rigor to these experiments and have altered our methods section to reflect this (see lines 740 – 741, 753 – 755, 757 – 759 and 760 – 764).

We thank the reviewer for their suggestion of counting 'puncta per ONL area' to improve the robustness of our quantification. We have done so, again using an automated macro in Imagej. Briefly, photoreceptor inner segment area was manually defined as the region of interest (ROI) for each image. Max intensity projections of the relevant channel were filtered for noise using 'Despeckle', then the threshold was adjusted to create a binary image with threshold levels set to allow identification of individual puncta (same threshold used for all images) and puncta were counted within the defined ROI using Analyse Particles. The results show similar, significant increases in p62, LAMP1 and VDAC1 expression in mutant retinas. We have updated Figure 3 and Figure 5 as a result, using this quantification method and updated the methods section to reflect this (see lines 855 – 864).

In addition to our immunofluorescence quantification, we observed increased p62 expression in both *Rpgr*^{Ex3d8} and *Pde6b*^{atrd2} mutants by western blotting and have added the data to Figure 3 and Figure 7. However, we prefer using immunofluorescence for our quantification as it allowed us to determine the sub-cellular localisation within photoreceptors of these proteins; important spatial data that would have been lost through western blotting of retinal lysates. In the case of VDAC1, however, we also see increased transcript expression in our mutant rods on scRNASeq.

(#7) Figure 6 and 7: quantification of pMLKL+ cells. Please use a similar method to document results for these two experiments. Here pMLKL+ cells per section (Figure 6D) or per μm^2 (Figure 7B) have been quantified. On the one hand, we do not know how large a “section” is – my advice is to convert Figure 6D to μm^2 . And on the other, in Figure 7b the numbers are crazily small – perhaps convert to something more meaningful (eg 100 μm^2 , mm²).

RESPONSE #7 We thank the reviewer for suggesting we use the same methods in our pMLKL quantification in Fig 6d and 7b. We have now done so, converting Fig 6d and 7b to cells/ 100 μm^2 to ensure uniformity.

Reviewer #2:

(Major #1) The evaluation of two transgenic strains harbouring mutations in the same gene is rather a poor approach to identify common altered pathway to be therapeutically targeted in a pathology, i.e. inherited retinal diseases, for which thousands of variants and hundreds of genes have been identified. At the end of the manuscript the authors briefly analyse -only at cellular level- the *Pde6Batrd2* mouse model and describe some phenotypic traits also observed in *Rpgr* strains. However, as stated before, alterations in autophagy, cell death and mitochondrial damage are already well known common hallmark of IRDs, therefore the author should provide a better molecular characterization of *Pde6Batrd2* to strengthen their conclusions and the novelty of the overall work.

MAJOR RESPONSE #1 We thank the reviewer for this comment and have spent time better characterising our *Pde6b^{atrd2}* line molecularly. These results can now be found in our edited Figure 7 and the new Figure 8. Our new data demonstrates:

- Increased levels of phosphorylated AKT in the *Pde6b^{atrd2}* mouse, similar to that seen in our *Rpgr^{Ex3d8}* line;
- Increased VDAC1 staining in the *Pde6b^{atrd2}* mouse, similar to that seen in our *Rpgr^{Ex3d8}* line;
- Quantification of autophagosomes noted on TEM show significant increase in numbers in the *Pde6b^{atrd2}* mouse.
- We also observed increased expression of OXPHOS pathway proteins Cytochrome C and the Complex IV protein, MtCO1 in the *Pde6b^{atrd2}* mouse (described in Major Response #3 below and shown in Supplementary Figure 5).

We hope that this new data, coupled with that in the original submission, will help strengthen our conclusions and the novelty of the work.

(Major #2) The authors postulated that ROS may promote the activation of necroptosis and trigger a “ROS induced by ROS” cycle which further increments cell death (lines 360-366). I suggest a careful evaluation of ROS levels in the three mouse models included in the manuscript – e.g. by using ex-vivo retinal explants- to strengthen this hypothesis.

MAJOR RESPONSE #2 Menger et al (2023) recently performed elegant mass spectrometry studies using the mitochondria- targeted mass spectrometric probe, MitoB, to assess for ROS production in RP mouse models.¹ They found that increased ROS production preceded photoreceptor death in both *Rpgr* and *Pde6b* mutant models. We have therefore, respectfully, not performed the reviewer’s excellent suggestion to assess for ROS production as we feel these studies are comprehensive and instead have referenced the paper in our discussion (Lines 644 – 645).

(Major #3) Many details concerning the assessment of the mitochondrial physiology are lacking, such as the expression pattern of OXPHOS proteins, the enzymatic activity of the respiratory complex, ATP production and membrane potential.’

MAJOR RESPONSE #3 The mitochondria field is currently limited in their ability to perform *in vivo* or *ex vivo* assessment of mitochondrial membrane potential ($\Delta\psi_m$) or ATP production. JC-1 dye assays ($\Delta\psi_m$) and ATP production assays are unfortunately not designed for tissue analysis but rather cell cultures. This has been overcome recently with the development of *in*

Medical Research Council Human Genetics Unit at the University of Edinburgh

Institute of Genetics and Cancer, Western General Hospital, Crewe Road, Edinburgh EH4 2XU

T: +44 (0)131 651 8500 ed.ac.uk/mrc-human-genetics-unit MRC_HGU

vivo probes like MitoClick², but they would require expertise of the technology and considerable time to optimise for use in the eye. We therefore, respectfully, argue that this is beyond the scope of this paper.

To further our characterisation of mitochondrial physiology in our models (and as suggested by Reviewer 2), we have assessed expression patterns of proteins involved in the OXPHOS pathway. Analysis shows increased expression of Cytochrome C and the Complex IV protein, MtCO1 in both *Rpgr*^{Ex3d8} and *Pde6b*^{atrd2} lines, compared to wild type littermate controls, with no differences observed in expression of complex I and III proteins (see lines 269 – 275 in the revised manuscript and ‘**Supplementary Figure 5. Mitochondrial biogenesis and oxidative phosphorylation are up-regulated in *Rpgr*^{Ex3d8} and *Pde6b*^{atrd2} mutants.**’) We feel this strengthens the evidence that mitochondrial dysfunction precedes rod death in both our models.

(Major #4) All the mechanistic hypothesis presented to explain the molecular basis giving rise to the cellular phenotype are based on the scRNAseq, whereas the proteins encoded by the differentially expressed genes are poorly characterized (e.g. total levels, changes in post-translational modifications etc). That is pivotal, due to the amount of post-translational modifications regulating the pathways that the authors propose to be involved in IRD pathogenesis. Moreover, at technical level, western blots may provide more reliable and quantifiable data in comparison to immunohistochemistry.

RESPONSE #4 Reviewer 2 feels the molecular evidence we provide to explain the cellular phenotypes seen in our mouse models is all based on our scRNAseq data, whilst characterisation of the translated proteins is lacking. Whilst these unbiased datasets provided our entry point to better defining pathway disruptions, we feel our immunohistochemistry experiments provide good supporting evidence (VDAC1 expression for mitochondrial stress, p62 and LAMP1 for autophagy dysfunction, pMLKL for necroptosis).

We agree that immunoblotting would help support the conclusions we make in this paper regarding expression of VDAC1 and p62, which we believe Reviewer 2 is referring to. We also agree that further analysis of post-translational modifications of these proteins would strengthen the conclusions regarding the pathways upregulated in our IRD models. To this end, we carried out western blots for phospho-SQSTM1/ phospho-p62 (S403)(Cell Signaling Technology cat# 39786). Phosphorylation of p62 at S403 increases the affinity of p62 for polyubiquitinated chains and is essential for its function in autophagic clearance (Matsumoto et al., 2011)³. We observed an increase in phospho-p62 in *Rpgr*^{Ex3d8} and *Pde6b*^{atrd2} mutant retinas as well as an overall increase in p62 (shown in revised Figure 3 and Figure 7), suggesting that the accumulating p62 in mutant photoreceptors is likely to be involved in mediating autophagy, as opposed to its role as a scaffold protein for other signalling pathways.

Reviewer 2 feels that, at a technical level, western blots provide more reliable quantitative data than immunohistochemistry. We, respectfully, argue that this is not always the case, with our photoreceptor VDAC1 experiments being an example. We believe our immunohistochemistry experiments, complete with quantification, support that there is upregulation in VDAC1 expression in the photoreceptor inner segments of our mutant lines (see **Figure 5a-c**), which is in keeping with the manuscript’s other data that shows mitochondrial structural and functional changes in the photoreceptor. This important spatial data is not captured by immunoblotting. Indeed, we performed western blotting of retinal lysates for VDAC1, which showed no detectable difference between mutant and wild type (see ‘**Rebuttal Figure 3. Western blots of VDAC1.**’). Examining Figure 5a-b, the large amount of VDAC1 expression in the outer plexiform layer is evident and could mask

Medical Research Council Human Genetics Unit at the University of Edinburgh

Institute of Genetics and Cancer, Western General Hospital, Crewe Road, Edinburgh EH4 2XU

T: +44 (0)131 651 8500 ed.ac.uk/mrc-human-genetics-unit  MRC_HGU

8

changes in levels in mutant IS. We therefore argue that, in this case, immunohistochemistry is the better method of assessment.

(Minor #1) In the Figure 2 a marked increase in mitophagy is clearly detectable: a great amount of autophagosomes are engulfing mitochondria. This aspect is very interesting and should be further studied (e.g. by performing quantifications here and in Figure 4 as well, by checking possible alterations in the expression of genes specifically involved in mitophagy).

RESPONSE Minor #1 We thank the reviewer for this interesting observation. Given the mitochondrial phenotype identified in our models, a resulting increase in mitophagy could explain our observations in Figures 2 and 3. As suggested, we have attempted to quantify the number of engulfed mitochondria observed in our TEM data that captures the distal inner segments. To do so, we have counted all autophagosomes containing membranous material. Our results suggest that there is, indeed, a significant increase in our mutant models of these membrane-containing autophagosomes (see '**Rebuttal Figure . Assessment of membrane-containing autophagosomes.**'). However, on discussion with the mitophagy expert Prof Ian Ganley (MRC Protein Phosphorylation and Ubiquitination Unit, University of Dundee), we have concerns of the robustness of this data. In essence, there is no guarantee that the observed membranes are truly mitochondria instead of, for example, engulfed endoplasmic reticulum. Further, analysis of mitophagy-related gene expression in our scRNASeq datasets show conflicting results between each line (see '**Rebuttal Figure 5. scRNASeq analysis of mitophagy-related genes.**'). We therefore do not feel confident enough to include this data in the manuscript. Ultimately, use of one of the new mitophagy reporter mouse strains⁴ would answer this really interesting question, but this is a significant project in itself and, we feel, beyond the scope of this paper.

(Minor #2) Alterations in general autophagy and mitophagy are frequently associated with alterations in mitochondrial biogenesis which is used to compensate the loss of dysfunctional mitochondria and repopulate the mitochondrial pool with newborn organelles. It would be interesting to check if master genes/proteins regulating mitochondrial biogenesis are altered in these models and go further deep in their molecular characterization.

RESPONSE Minor #2 We thank Reviewer 2 for this suggestion and have re-analysed our scRNASeq datasets. Results support Reviewer 2's hypothesis; mitochondrial biogenesis genes are, indeed, upregulated in both datasets. We include these as an extra supplementary figure (see '**Supplementary Figure5. Mitochondrial biogenesis and oxidative phosphorylation are up-regulated in Rpgr^{Ex3d8} and Pde6b^{atrd2} mutants.**') and referred to this in the manuscript in lines 267 – 269.

(Minor #3) In lines 89-92 authors observed that "Immunofluorescent staining of retina cryosections confirmed cells, positive for the pan-macrophage marker F4/80 and the microglial marker TMEM119, invading outer layers of RpgrORFd5 and RpgrEx3d8 retinas at 18 months, something not observed in wild type at this age (Supplementary Figs. 2e, 3e)". I cannot see any F4/80 or TMEM119 positive cell in the Supplementary Fig. 3e and that would fit with the scRNAseq data (line 95-97). Therefore, the authors should discuss about this difference between the two strains, given that inflammation is a phenotype very linked to neurodegeneration.

RESPONSE Minor #3 We have replaced the original images in Supplementary Figures. 2e, 3e with new ones using P2Y12 as an alternative marker of resident microglia which labels these cells more clearly than TMEM119. These images, we hope, now show more clearly

Medical Research Council Human Genetics Unit at the University of Edinburgh

Institute of Genetics and Cancer, Western General Hospital, Crewe Road, Edinburgh EH4 2XU

T: +44 (0)131 651 8500 ed.ac.uk/mrc-human-genetics-unit MRC_HGU

the F4/80 / P2Y12 positive cells in the mutant outer nuclear layer.

(Minor #4) On the same page of some mayor comment, I miss western blot analysis in Figure 5 not only for VDAC, that is a structural protein, but also for proteins involved in mitochondrial function (such as OXPHOS proteins). The latter will significantly strengthen the message.

RESPONSE Minor #4 As discussed in Major response **#3**, we have assessed expression patterns of OXPHOS proteins. Analysis shows increased expression of Cytochrome C and the Complex IV protein, MtCO1 in both *Rpgr*^{Ex3d8} and *Pde6b*^{atrd2} lines, compared to wild type littermate controls, with no differences observed in expression of complex I and III proteins. We include these is our updated manuscript as '**Supplementary Figure 5. Mitochondrial biogenesis and oxidative phosphorylation are up-regulated in *Rpgr*^{Ex3d8} and *Pde6b*^{atrd2} mutants**' and refer to this in lines 269 – 275 of the revised manuscript. We agree with Reviewer 2 that this now significantly strengthens the message.

As discussed in Major response **#4**, we have performed western blotting of VDAC1, which shows no significant change in expression in retinal lysates (see '**Rebuttal Figure 3. Western blots of VDAC1.**'). However, as argued above, we feel this is not the appropriate experiment for characterising photoreceptor VDAC1 expression, given the high levels seen in the outer plexiform layer. Immunohistochemistry allows us to determine the significant expression changes at a sub cellular level, whereas immunoblotting misses this important spatial data.

(Format/stylistic #1) I miss an introductory section on *Rpgr* and the encoded proteins (acronym, function, presence of retina-specific isoforms etc.) and, eventually, a schematic representation of the mutant proteins.

RESPONSE Format/stylistic #1 We have added to the manuscript descriptions of the acronym and function of RPGR and discussed the retinal specific isoform (see lines 67 - 68 of revised manuscript). We have referred readers to a review of RPGR from our group (Megaw et al., 2015) for further information. We are concerned at the expanded number of figures in our manuscript following revisions and have instead referred readers to the schematic representation of the mutant proteins in the original paper that describes the mice (Megaw et al., 2024). We hope this is satisfactory.

(Format/stylistic #2) The age of the animals is not always indicated in the figure legend (e.g. Figure 3).

RESPONSE Format/stylistic #2 We apologise for this and have amended the manuscript to include all animal ages.

(Format/stylistic #3) Check that all the figures in the main body of the text are called out in sequential order (e.g. "(Fig.2c)", line 194 or "Supplementary Fig.4c", line 257)

RESPONSE Format/stylistic #3 We apologise for this and have amended the manuscript to ensure sequential order of our figures when they are first called out is observed. Of note, we believe the specific examples the reviewer gives had been previously referred to earlier in the manuscript; Fig 2c (originally line 194, now 259 in revised manuscript), is initially

referenced on line 186 in revised manuscript; Supplementary Fig 4c (originally line 257; now 393 in revised manuscript) is initially referenced on line 135 in revised manuscript.

(Format/stylistic #4) Complete the information regarding the “n” used for statistics, by specifying the number of sections and/or retinas and/or animals. This should usually be included in each figure legend.

RESPONSE Format/stylistic #4 We apologise for this and have amended the figure legends to ensure these are included.

References

1. Menger KE, Logan A, Luhmann UFO, Smith AJ, Wright AF, Ali RR, Murphy MP. *In vivo* measurement of mitochondrial ROS production in mouse models of photoreceptor degeneration. *Redox Biochem Chem*. 2023 Dec;5-6:None. doi: 10.1016/j.rbc.2023.100007. PMID: 38046619; PMCID: PMC10686909.
2. Logan A, Pell VR, Shaffer KJ, Evans C, Stanley NJ, Robb EL, Prime TA, Chouchani ET, Cochemé HM, Fearnley IM, Vidoni S, James AM, Porteous CM, Partridge L, Krieg T, Smith RA, Murphy MP. Assessing the Mitochondrial Membrane Potential in Cells and In Vivo using Targeted Click Chemistry and Mass Spectrometry. *Cell Metab*. 2016 Feb 9;23(2):379-85. doi: 10.1016/j.cmet.2015.11.014. Epub 2015 Dec 17. PMID: 26712463; PMCID: PMC4752821.
3. Matsumoto G, Wada K, Okuno M, Kurosawa M, Nukina N. Serine 403 phosphorylation of p62/SQSTM1 regulates selective autophagic clearance of ubiquitinated proteins. *Mol Cell*. 2011 Oct 21;44(2):279-89. doi: 10.1016/j.molcel.2011.07.039
4. McWilliams TG, Prescott AR, Allen GF, Tamjar J, Munson MJ, Thomson C, Muqit MM, Ganley IG. mito-QC illuminates mitophagy and mitochondrial architecture in vivo. *J Cell Biol*. 2016 Aug 1;214(3):333-45. doi: 10.1083/jcb.201603039. Epub 2016 Jul 25. PMID: 27458135; PMCID: PMC4970326.